



# Towards improved turbulence estimation with Doppler wind lidar VAD scans

Norman Wildmann[1], Eileen Päschke[2], Anke Roiger[1], and Christian Mallaun[3]

[1]Deutsches Zentrum für Luft- und Raumfahrt e.V., Institut für Physik der Atmosphäre, Oberpfaffenhofen, Germany
[2]DWD, Meteorologisches Observatorium Lindenberg - Richard-Aßmann-Observatorium, Lindenberg, Germany
[3]Deutsches Zentrum für Luft- und Raumfahrt e.V., Flugexperimente, Oberpfaffenhofen, Germany

**Correspondence:** Norman Wildmann (norman.wildmann@dlr.de)

**Abstract.** The retrieval of turbulence parameters with profiling Doppler wind lidars (DWL) is of high interest for boundary-layer meteorology and its applications. The DWL measurements extend beyond the observations with meteorological masts and are comparably flexible in their installation. Velocity-azimuth display (VAD) type scans can be used to retrieve turbulence kinetic energy (TKE) dissipation rate through a fit of measured azimuth structure functions to a theoretical model. At the
elevation angle of 35.3° it is also possible to derive TKE. We show in this study how modifications to existing methods allow to retrieve TKE and its dissipation rate even with a small number of scans, how a simple correction for advection improves the results at low altitudes and that VAD scans at different elevation angles with the same instrument provide comparable results of TKE dissipation rate after all filters and corrections. For this purpose, data of two experiments are utilized: First, measurements at the Observatory Lindenberg – Richard-Aßmann Observatory (MOL-RAO) are used for validation of the DWL retrieval with
sonic anemometers on a meteorological mast. Second, distributed measurements of three DWL during the CoMet campaign are analyzed and compared to in-situ measurements of the DLR Cessna Grand Caravan 208B. The comparison to in-situ instruments shows that the methods to improve turbulence retrievals from VAD scans introduced in this study are effective, especially at low altitudes and for narrow cone angles, but it also shows the limits of turbulence measurement with state-of-the-art DWL in low turbulence regimes.

# 1  Introduction

The observation of turbulence in the atmosphere and in particular the atmospheric boundary layer (ABL) is of great importance for basic research in boundary-layer meteorology as well as in applied fields such as aviation, wind energy (van Kuik et al., 2016; Veers et al., 2019) or pollution dispersion (Holtslag et al., 1986).

A wide range of instruments are used to measure turbulence: sonic anemometers are nowadays the most popular in-situ in-
strument which can be installed on meteorological masts and provide continuous data of the three-dimensional flow and its turbulent fluctuations (Liu et al., 2001; Beyrich et al., 2006). For in-situ measurements above the height of towers, airborne systems are applied such as manned aircraft (Bange et al., 2002; Mallaun et al., 2015), remotely-piloted aircraft systems (RPAS, van den Kroonenberg et al., 2011; Wildmann et al., 2015) or tethered lifting systems (TLS, Frehlich et al., 2003) which can be equipped with turbulence probes such as multi-hole probes or hot wire anemometers. A different category of instruments are





remote-sensing instruments such as radar, sodar and lidar which can measure wind speeds and allow the retrieval of turbulence based on assumptions of the state of the atmosphere and the structure of turbulence. In this study, we focus on ground-based Doppler wind-lidars (DWL), which have become increasingly popular in boundary-layer research because of their light weight, invisible and eye-safe lasers, their reliability and high availability which is only restricted by clouds/fog and rain or very low

aerosol content in the atmosphere.

A variety of methods already exist to retrieve turbulence from DWL measurements. They can be categorized according to the respective scanning strategy applied: the simplest scanning pattern is a constant vertical stare to zenith, which allows to obtain variances of vertical velocity and estimates of turbulence kinetic energy (TKE) dissipation rate (O'Connor et al., 2010; Bodini et al., 2018). More complex are conical scans (velocity azimuth display, VAD) with continuous measurements along the cone

(Banakh et al., 1999; Smalikho, 2003; Krishnamurthy et al., 2011; Smalikho and Banakh, 2017). These scans include information on the horizontal wind component as well. A simplification of VAD scans are Doppler-beam-swinging (DBS) methods, that reduce the number of measurements taken along the cone to a minimum of 4-5 beams and thus increase the update rate for single wind profile estimations (Kumer et al., 2016). Both, VAD and DBS are popular scanning strategies that are applied in commercial instruments. Kelberlau and Mann (2019a, b) introduced new methods to obtain better turbulence spectra from

conically scanning lidars by corrections for the scanner movement. Significantly different scanning strategies are vertical (or horizontal) scans which can also provide vertical profiles of turbulence (Smalikho et al., 2005), but even allow deriving two-dimensional fields of TKE dissipation rate (Wildmann et al., 2019). Multi-Doppler measurements require more than one lidar with intersecting beams, but do not need assumptions on homogeneity to measure turbulence at the points of the intersection directly (Fuertes et al., 2014; Pauscher et al., 2016; Wildmann et al., 2018). For operational or continuous monitoring of vertical

profiles of turbulence in the ABL, VAD or DBS scans are most suitable. At an elevation angle of $35.3°$, a VAD scan allows to retrieve TKE, its dissipation rate, integral length scale and momentum fluxes according to a method that was first developed for radar by Kropfli (1986) and adapted for lidar later by Eberhard et al. (1989) using the variance of radial velocities along the scanning cone. Further improvements of this method have been implemented by Smalikho and Banakh (2017) and Stephan et al. (2018), which also account for lidar volume averaging effects. For conditions with significant advection, the method is

not applicable and causes large errors, especially at low altitudes where the cone diameter of the VAD scan is small. With this study, we propose a method to significantly reduce this error. We also apply the turbulence retrieval to VAD scans with $75°$ elevation angle, which still allows to retrieve TKE dissipation rate. In this case the advection correction is particularly important. The experiments that were carried out are explained in Sect. 2. The methods and the new developments are explained in Sect.3. A focus of this study is on the validation of the lidar measurements with sonic anemometers and airborne in-situ

measurements. The results of the validation are presented in Sect. 4. Conclusions and an outlook are given in Sect. 5.

## 2  Experiment description

In this study, data from two different sites and sets of instruments are analyzed. Both of the sites and the instrumentation is introduced in this Section.



## 2.1 The MOL-RAO Falkenberg field site

The Meteorological Observatory Lindenberg – Richard-Aßmann Observatory (MOL-RAO) is part of Deutscher Wetterdienst (DWD), the national meteorological service of Germany. The observatory is situated in the East of Germany, approximately 65 km to the South-East of the center of Berlin. MOL-RAO runs a comprehensive operational measurement program to charac-

terize the physical structure and processes in the atmospheric column above Lindenberg. Measurements of ABL processes form an essential part of it, they are carried out at the boundary layer field site (in German: Grenzschichtmessfeld, GM) Falkenberg, about 5 km to the South of the main observatory site. The GM Falkenberg is situated in a rural landscape dominated by forest, grassland and agricultural fields (see Fig. 1). A central measurement facility at the Falkenberg site is a 99m tower, equipped with booms to carry sensors every 10 m.

Since 2014, MOL-RAO is using a DWL „Stream Line" (Halo Photonics Ltd.) for boundary layer measurements. From that time the device has been extensively tested with respect to its operational use for wind and turbulence measurements. This included, for instance, tests on the technical robustness and data availability under all weather conditions, but also tests of different scanning strategies and retrieval methods for the 3D wind vector and for the TKE. The position of the DWL during a measurement period from 2 April 2019 through 30 April 2019 was at the western edge of the field site, at about 500 m distance

from the 99m tower. It should be noted that there is a small patch of forest about 300 m to the W-NW of the lidar site. During this period the system continuously performed VAD scans with an elevation angle of 35° elevation which will be analysed for turbulence retrievals in this study.

Continuous turbulence measurements (20 Hz sampling frequency) using the eddy-covariance method with sonic anemometer-thermometers USA-1 (METEK GmbH) and infrared gas analysers LI7500 (LiCor Inc.) are performed at the 50m and 90m

levels of the tower and have been used for validation purposes. The instruments are mounted at the tip of the booms pointing towards South with the LI7500 behind the USA-1.

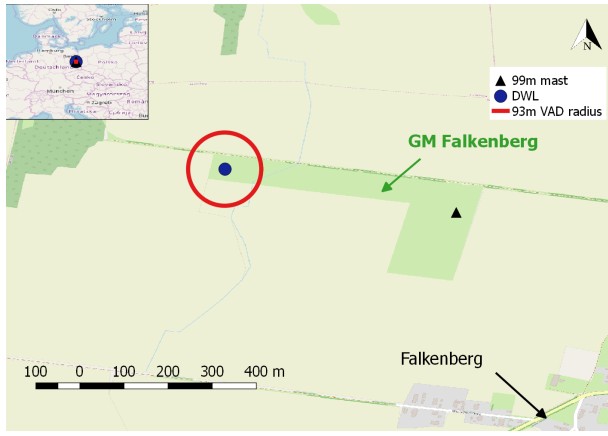

**Figure 1.** Sketch of the measurement site at MOL-RAO, GM Falkenberg. Map data ©OpenStreetMap contributors 2019. Distributed under a Creative Commons BY-SA License.



## 2.2 The CoMet (CO$_2$ and Methane) Mission 2018

Within the scope of the CO$_2$ and Methane Mission (CoMet) that was conducted in spring 2018, three Doppler wind lidars of type Leosphere Windcube 200S (details see Tab. 1) were installed in Upper Silesia with the purpose of providing spatially distributed wind and turbulence measurements in the ABL. CoMet aims at a better understanding of the budgets of the two

most important anthropogenic greenhouse gases, CO$_2$ and CH$_4$. For this purpose, the research aircraft HALO (high altitude and long range) was taking remote sensing and in-situ measurements over large parts of the European continent. A dedicated area of high interest was the region of Upper Silesia, where large amounts of methane are known to be released due to the intensive coal extraction activities in the respective Coal Basin. During the CoMet campaign, the DLR Cessna Grand Caravan 208B (D-FDLR) aircraft was equipped with in-situ instruments to measure greenhouse gases as well as thermodynamic variables.

The DWL measurements are particularly helpful to support the CoMet measurements by providing wind information which is essential to derive emission flux estimates from passive remote sensing (Luther et al., 2019) or in-situ measurements of mass concentrations (Fiehn et al., 2020). The DWL wind information can also be used to validate modeled wind of the transport models for greenhouse gases. The lidars were remotely operated during the whole CoMet campaign period from 16 May 2017 to 17 June 2017 and were continuously measuring. The locations of the three lidars were planned to cover the whole region of

interest and were finally fixed based on logistical constraints.

The lidars were operating in VAD modes with two different elevation angles. Since the focus for the CoMet campaign was on continuous wind profiling and a good height coverage was desired, the lidars were programmed to perform VADs with an elevation angle of 75° (see Tab. 1, VAD75) for a longer period, i.e. 24 scans (≈29 minutes), followed by only six scans (≈7 minutes) at 35° elevation (VAD35) for turbulence retrievals.

As shown in Fig. 2, the three lidars were separated by several tens of kilometers and are located in different terrain types. While DLR#1 is in a mixed rural and urban area, DLR#2 is in a mostly forested environment and DLR#3 is in close vicinity to the lake Goczalkowicki. The main wind direction during the campaign was from the East, with particularly strong winds during nighttime low-level jet (LLJ) events. In this study we analyze statistics of the whole campaign, as well as a case study on 5 June 2017, on which D-FDLR was performing long straight and level legs between 800 m and 1600 m as indicated in the flight

path in Fig. 2.



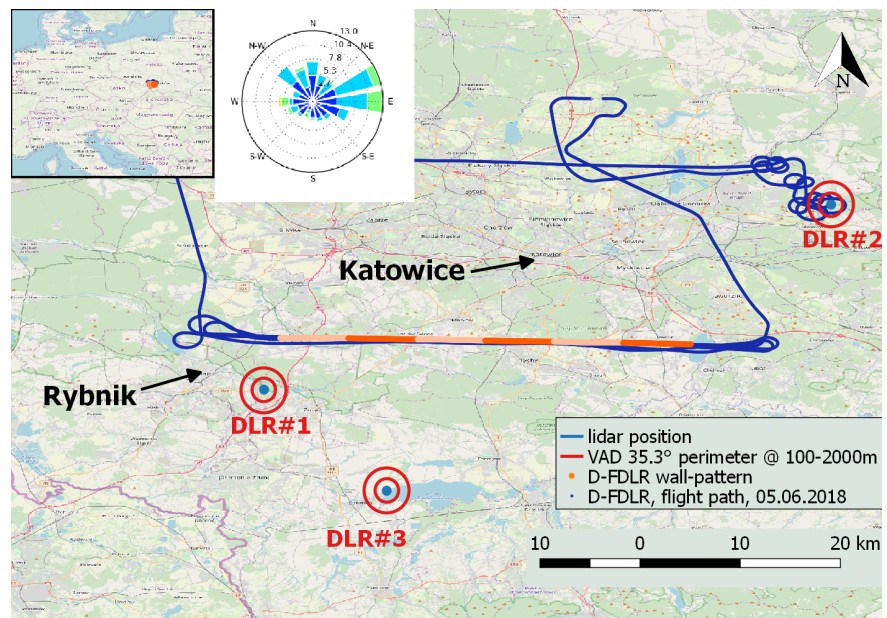

**Figure 2.** Sketch of the measurement site in Upper Silesia. Red circles show the extent of the VAD-scan at 35.3° for 100 m and 2000 m at the respective lidar location. The orange line marks the flight path of D-FDLR on 5 June 2017. The different shades of orange are used to indicate a subdivision of the flight leg in shorter sublegs. Map data ©OpenStreetMap contributors 2019. Distributed under a Creative Commons BY-SA License.

**Table 1.** Main technical specifications of the Doppler wind lidars.

|  | Windcube 200S, VAD75 | Windcube 200S, VAD35 | Stream Line |
|---|---|---|---|
| Wavelength $\lambda$ | 1.54 µm | 1.54 µm | 1.5 µm |
| Pulse length $\tau_p$ | 200 ns | 200 ns | 180 ns |
| Time window $T_w$ | 288 ns | 144 ns | 240 ns |
| Bandwidth | 26.7 m s$^{-1}$ | 26.7 m s$^{-1}$ | 19.4 m s$^{-1}$ |
| Elevation angle $\varphi$ | 75° | 35.3° | 35.3° |
| Angular speed | 5° s$^{-1}$ | 5° s$^{-1}$ | 5° s$^{-1}$ |
| Pulse repetition frequency | 20 kHz | 20 kHz | 15 kHz |
| Accumulation time | 200 ms | 200 ms | 133 ms |
| CNR filter | -20..0 dB | -20..0 dB | -15..0 dB |





## 3 Methods

### 3.1 Sonic anemometer turbulence measurements

From the sonic anemometers on the meteorological mast, TKE and TKE dissipation rate are calculated. TKE is calculated from the sum of variances $E_{\mathrm{TKE}} = 0.5 \left( \sigma_u^2 + \sigma_v^2 + \sigma_w^2 \right)$. TKE dissipation rate $\varepsilon$ is estimated through a fit of the measured

second-order structure function of horizontal velocity to the theoretical, longitudinal Kolmogorov-structure function in the range $\tau_1 = 0.1$ s to $\tau_2 = 2$ s. As in Muñoz-Esparza et al. (2018), who showed that the structure function method is more robust than estimates from spectra, the values are calculated for 2-minute intervals and then averaged over half-hour periods. The geometry of the sonic anemometer setup disturbs the measurements for wind directions from 330° to 50° (see also App. D). Data for these wind directions are removed from the analysis.

### 3.2 VAD turbulence measurements

Methods to retrieve turbulence parameters from VAD scans are well-known and a variety of different methods exist. The method we refine in this study is based on the theory that was originally described by Eberhard et al. (1989) for lidar measurements. The variance of radial velocities $\sigma_r^2$ depends on the range gate distance $R$, the azimuth angle $\theta$ and the elevation angle $\varphi$. It is calculated from the measured radial wind speeds $V_r$:

$$v_r(R,\theta,\varphi,t) = V_r(R,\theta,\varphi,t) - \langle V_r(R,\theta,\varphi) \rangle \tag{1}$$
$$\sigma_r^2 = \langle v_r(R,\theta,\varphi,t)^2 \rangle \tag{2}$$

From a partial Fourier decomposition (see App. A) and for the special case of $\varphi = 35.3°$ a simple equation for $E_{\mathrm{TKE}}$ is derived:

$$E_{\mathrm{TKE}} = \frac{3}{2}\overline{\sigma}_r^2 \quad . \tag{3}$$

In this equation, $\overline{\sigma}_r^2$ is the mean of the variance of radial velocites over all azimuth angles. In the following, we will refer to this method as E89-retrieval.

In order to retrieve estimations of TKE dissipation rate $\varepsilon$ from VAD scans, a similar approach to the method for sonic anemometers can be followed. A fit of the azimuth structure function to the equation

$$D_r(\psi) = (4/3)C_K(\varepsilon \psi R')^{2/3} \quad , \tag{4}$$

with $D_r$ the transverse structure function of radial velocities, $C_K$ the Kolmogorov constant, $\psi$ the azimuth angle increment and $R' = R\cos\varphi$ retrieves an estimate for $\varepsilon$ according to Smalikho and Banakh (2017). We will refer to this method as S17A in the following.





Scanning with Doppler lidar in a VAD implies a volume averaging of radial velocities in longitudinal and transversal direction. The E89 and S17A methods do not consider this effect and will thus yield a systematic underestimation of TKE and $\varepsilon$. Smalikho and Banakh (2013) proposed a theory that considers the volume averaging and allows the retrieval of $\varepsilon$ from conical scans, independent of the elevation angle. In Smalikho and Banakh (2017), this method has been combined with the E89-

5 method to yield TKE, $\varepsilon$ and the momentum fluxes. It is based on the decomposition of radial wind speed variance $\sigma_r^2$ into its subcomponents, i.e. $\sigma_L^2$ as the lidar measured variance, $\sigma_a^2$ as the lidar measured variance without instrumental error $\sigma_e^2$, and the turbulent broadening of the lidar measurement $\sigma_t^2$. In Smalikho and Banakh (2017), all of these variances and structure functions are calculated for single azimuth angles and then averaged. We describe in Sect.3.2.1 why we use total variances and structure functions of all radial velocities.

10 $$\sigma_L^2 = \sigma_a^2 + \sigma_e^2 \tag{5}$$

$$\sigma_a^2 = \sigma_r^2 - \sigma_t^2 \tag{6}$$

$$\sigma_r^2 = \sigma_L^2 + \sigma_t^2 - \sigma_e^2 \tag{7}$$

Additionally, the measured azimuth structure function $D_a(\psi_l)$ (as a function of separation angle $\psi_l$, where $l$ is the index of the discrete separation angle of the scan) can be decomposed into the lidar measured structure function and the instrumental 15 error:

$$D_a(\psi_l) = D_L(\psi_l) - 2\sigma_e^2 \tag{8}$$

Substituting $\sigma_e^2$ in Eq. 7 with Eq. 8 yields:

$$\sigma_r^2 = \sigma_L^2 + \sigma_t^2 - \frac{1}{2}D_L(\psi_l) + \frac{1}{2}D_a(\psi_l) \quad . \tag{9}$$

With $\langle v_r^2 \rangle = \sigma_r^2$ in Eq.3, TKE can be redefined as a function of the measured line of sight variances $\sigma_L^2$, the measured 20 lidar azimuth structure function of radial velocities $D_L(\psi_1)$ and a residual term $G$, which includes the two unknowns $\sigma_t^2$ and $D_a(\psi_1)$:

$$E_{\text{TKE}} = \frac{3}{2}\left[\sigma_L^2 - \frac{D_L(\psi_1)}{2} + G\right] \quad , \tag{10}$$

$$G = \sigma_t^2 + \frac{1}{2}D_a(\psi_1) \tag{11}$$

In Banakh and Smalikho (2013), a relationship between the two unknowns and TKE dissipation rate is theoretically derived 25 from the two-dimensional Kolmogorov-Obhukov spectrum as

$$\sigma_t = \varepsilon^{2/3}F(\Delta y) \tag{12}$$

$$D_a(\psi_l) = \varepsilon^{2/3}A(l\Delta y) \quad , \tag{13}$$





where $F(\Delta y)$ and $A(l\Delta y)$ are model functions that include the lidar filter functions (see App. B). The lidar filter functions in longitudinal direction depend on pulse width of the laser beam $\Delta p$ and the time window $T_w$ of the data acquisition. The transverse filter function is defined by $\Delta y = R\Delta\theta\cos\varphi$, which is the distance the lidar beam moves along the cone during one accumulation period. The parameters for the lidars in this study are provided in Tab. 1 and are calculated from information

given by the manufacturer for the specific lidar type. Hence, $G$ depends on the turbulence dissipation rate $\varepsilon$:

$$G = \varepsilon^{2/3}\left[F(\Delta y) + \frac{A(\Delta y)}{2}\right] \tag{14}$$

From Eq. 13 and 8, $\varepsilon$ can be retrieved by the relation of measured differences in the structure function and the model structure function:

$$\varepsilon = \left[\frac{D_L(\psi_l) - D_L(\psi_1)}{A(l\Delta y) - A(\Delta y)}\right]^{3/2} \tag{15}$$

This equation does not depend on the elevation angle, so that the method allows the retrieval of $\varepsilon$ from VAD scans with other elevation angles as well. Figure 3 gives an example of the different structure functions that are calculated in this method (i.e. $D_L$, $D_a$ and $A$) and also gives a comparison to the structure function $D_s$ as calculated from sonic anemometer measurements. The value of $l = 9$ is chosen following the example of Smalikho and Banakh (2017) and corresponds to $l\Delta\theta = 9°$ as it was found to be suitable in all conditions in that study. In Fig. 3 the range that is thus used for the structure function fit is indicated

by the bold black line.

The retrieval method for $\varepsilon$ using Eq. 15 and TKE using Eq. 10 will be referred to as S17 in the following.

### 3.2.1    Modifications for small number of scans

The VAD at $\varphi = 35.3°$ during the CoMet-campaign was not run continuously, but only six individual scans are performed successively before switching back to the VAD at $\varphi = 75°$ as described in Sect. 2.2. This means that only six data points are

available to calculate variance and mean of the radial wind speeds at each azimuth angle, which cannot be considered a solid statistic. We introduce two modifications of data processing to overcome this problem which are based on the assumptions of stationary and homogeneous turbulence.

**Practical implementation of the ensemble average**

In Eq. 2, $\langle V_r(R,\theta,\varphi)\rangle$ can be calculated as the arithmetic mean of radial wind speeds at specific azimuth angles:

$$\langle V_r(R,\theta,\varphi)\rangle = \frac{1}{N}\sum_{n=0}^{N}V_r(R,\theta_n,\varphi)\quad, \tag{16}$$

where $N$ is the number of scans. Instead of this approach, we suggest to use the reconstructed radial wind speed from the retrieved wind field over all individual scans as the expected value in the variance calculation. For the retrieval of the three wind components $(\hat{u}, \hat{v}, \hat{w})$, filtered sine-wave fitting is applied (Smalikho, 2003). The reconstructed radial wind speeds $\hat{V}$ are

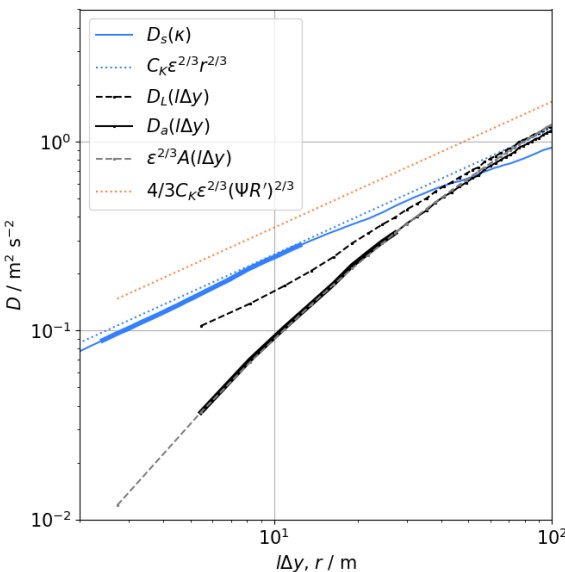

**Figure 3.** Example of structure functions of sonic anemometer (blue) and lidar (grey) at 90 m height on 4 April 2019, 1200-1230 UTC. The dashed black line shows the measured lidar structure function $D_L$, the solid black line $D_a$ is corrected for the systematic error $\sigma_e$ (see Eq. 8). The grey dashed line gives the model structure function $A$ and the dotted lines indicate the reconstructed inertial subrange for the calculated values of $\varepsilon$. The parts with bold lines are those ranges that are used for the structure function fits.

then used as the expected value in the variance calculation:

$$\hat{V} = \hat{w}(R)\sin\varphi + \hat{v}(R)\cos\varphi\cos\theta + \hat{u}(R)\cos\varphi\sin\theta \tag{17}$$

$$\langle V_r(R,\theta,\varphi)\rangle = \hat{V}(R,\theta,\varphi) \quad . \tag{18}$$

With this approach, all measurement points in the VAD with the same elevation angle are used to obtain the expected value $\langle V_r \rangle$ and thus, a better statistical significance is achieved. This method has also been proposed in Smalikho and Banakh (2017) as a practical implementation of Eq. 2.

**Averaging of variances**

In (Smalikho and Banakh, 2017), the variances of the lidar measurements are defined as the average of variances at individual azimuth angles:

$$\overline{\sigma}_r^2 = \frac{1}{M}\sum_{m=0}^{M}\sigma_r^2(\theta_m) \tag{19}$$





The variances $\sigma_r^2(\theta_m)$ are variances of a subsample of radial wind speeds of the VAD (i.e. those at a specific azimuth angle $\theta_m$). We use a simple relation between the variances of subsamples and the total variance of a dataset (see Appendix C). Applying this to the radial wind speed variances yields:

$$\sigma_r^2 = \frac{k-1}{n-1} \sum_{j=1}^{g} \sigma_r^2(\theta_m) + \frac{k(g-1)}{k-1} \overline{v}_r \quad , \tag{20}$$

where $k$ is the number of samples at each azimuthal angle, $g$ is the number of subsamples (here: $g = 360$ for all azimuth angles) and $n$ is the number of total samples in the dataset ($n = gk$). Since the mean of the radial wind speed fluctuations $\overline{v}_r = 0$ by definition, it is:

$$\sigma_r^2 \approx \overline{\sigma}_r^2(\theta_m) \quad . \tag{21}$$

### 3.2.2  Filtering of bad estimates

Improvements of turbulence estimates in low signal conditions can be achieved with filtering of bad estimates as described in Stephan et al. (2018). This approach is not based on the calculation of the azimuth structure function from measured radial wind speeds, but uses probability density functions (PDFs) and their corresponding standard deviations. The model PDF is defined as a Gaussian function with a filter term $P$:

$$p_M(x) = \frac{1-P}{\sqrt{2\pi}\sigma} \exp\left[-\frac{1}{2}\left(\frac{x}{\sigma}\right)^2\right] + \frac{P}{B_v} \quad , \tag{22}$$

where $P$ is the probability of bad estimates of $x$, $\sigma$ is the standard deviation of the PDF and $B_v$ is the velocity bandwidth of the lidar. Measured PDFs of the variables $v_r(R,\theta)$, $\Delta v_r(R,\theta + \Delta\theta)$ and $\Delta v_r(R,\theta + l\Delta\theta)$ are fit to the model PDFs to obtain an estimation of the corresponding standard deviations $\sigma_1$, $\sigma_2$ and $\sigma_3$ and probability of bad estimates $P_1$, $P_2$ and $P_3$. However, since the PDFs cannot be assumed Gaussian in atmospheric turbulence, the standard deviations are finally calculated as the integral over the measured PDFs in the range $\pm 3.5\sigma$ according to Stephan et al. (2018).

Replacing $\sigma_L^2$ with $\sigma_1^2$, $D_L(\psi_1)$ with $\sigma_2^2$ and $D_L(\psi_l)$ with $\sigma_3^2$ in Eqs. 10 and 15 yields:

$$E_{\mathrm{TKE}} = \frac{3}{2}\left[\sigma_1^2 - \frac{\sigma_2^2}{2} + G\right] \quad \text{and} \tag{23}$$

$$\epsilon = \left[\frac{\sigma_3^2 - \sigma_2^2}{A(l\Delta y) - A(\Delta y)}\right]^{3/2} \tag{24}$$

As suggested in Stephan et al. (2018), Eqs. 23 and 24 are only used if $P > 0$. We also introduced a quality control to discard 25    any measurements with $P > 0.5$ for best results. In practice, this method will thus only be applied in some conditions when the signal is weak and can extend the range of vertical profiles to some degree.





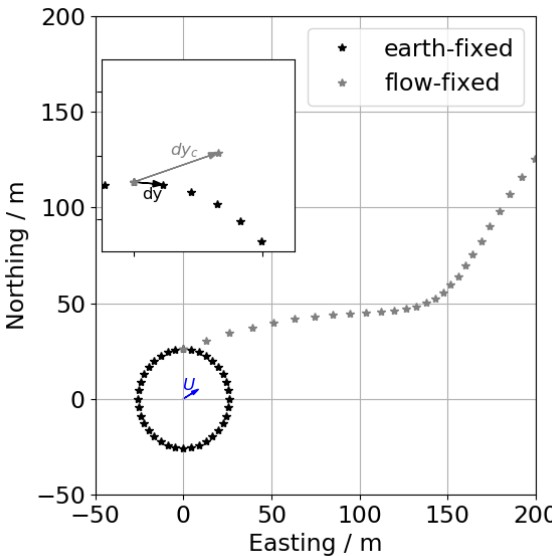

**Figure 4.** Sketch of measurement points of a VAD scan in an earth-fixed versus a flow-fixed coordinate system.

### 3.2.3 Correction for advection

The azimuth structure function and the volume average filter are distorted by advection through a modification of $\Delta y$. The effect is illustrated in Fig. 4. It shows that the distance between measurement points in a flow-fixed coordinate system is unequally spaced and on average larger than in the earth-fixed coordinate system. We propose a simplified correction as follows:

$$\Delta y = \Delta \theta R \cos \varphi \tag{25}$$

$$\Delta y \approx \frac{1}{N} \sum_{i=0}^{N} \sqrt{dx_i^2 + dy_i^2} \tag{26}$$

$$dx_i = x_{i+1} - x_i \tag{27}$$

$$dx_{c,i} = dx_i + \cos \Psi U \Delta t \tag{28}$$

$$dy_{c,i} = dy_i + \sin \Psi U \Delta t \tag{29}$$

$$\Delta y_c \approx \frac{1}{N} \sum_{i=0}^{N} \sqrt{dx_{c,i}^2 + dy_{c,i}^2} \quad . \tag{30}$$

Here, $R$ is the range gate distance, $\varphi$ is the elevation angle, $x_i$ and $y_i$ are the measurement point locations , $\Psi$ is wind direction, $U$ is wind speed and $\Delta t$ is the accumulation time of the lidar. The terms $\cos \Psi U \Delta t$ and $\sin \Psi U \Delta t$ describe the effect of advection on the measurement location in $x$- and $y$-direction respectively. Using the corrected measurement location displacements $dx_{c,i}$ and $dx_{c,i}$, we can calculate a corrected mean transverse sensing volume $\Delta y_c$. This method does not account for the unequal spacing, but corrects the average separation of data points, which is particularly important for the statistical evaluation





of turbulence.

The effects of advection on the turbulence estimation is largest in the lowest levels of the VAD-scans, because $\Delta y$ is small compared to $U\Delta t$ in this case. The retrieval method including the filtering for bad estimates, and the advection correction is referred to as W19 in the following.

### 3.2.4 Quality control filters

In order to fulfill the assumptions that are made with regards to the turbulence model and the turbulence retrieval method, the data is filtered according to the criteria given in Smalikho and Banakh (2017):

$$l\Delta y \ll L_v \quad , \tag{31}$$

$$L_v > \max\{\Delta z, \Delta y\} \tag{32}$$

$$R'\omega_s \gg |\langle \mathbf{V} \rangle| \tag{33}$$

For the purpose of evaluating the methods in a broad range, we set mild criteria for Eq. 31 and 33 using

$$l\Delta y < 2L_v \quad \text{and} \tag{34}$$

$$R'\omega_s > 2|\langle \mathbf{V} \rangle| \quad . \tag{35}$$

Equations 31 and 32 are criteria that require the integral length scale $L_v$ to be larger than the sensing volume of the lidar in
transversal ($\Delta y$) and longitudinal ($\Delta z$) direction. Unfortunately, there is no independent measurement of $L_v$ at all heights of the VAD scan, so that it is derived from the lidar measurement itself as $L_v = 0.3796\frac{E^{3/2}}{\varepsilon}$ (Smalikho and Banakh, 2017).

The filter criteria in Eq. 33 is a filter for conditions with significant advection which distorts the measured structure functions and is only applied if the method described in Sect.3.2.3 is not used.

Except for the retrieval method WS19, which uses the filtering of bad estimates, we set fixed CNR filter thresholds adapted to
the lidar type. Since the turbulence retrievals are very sensitive to bad estimates, we set the CNR thresholds to conservative values that are given in Tab. 1.

An overview of all retrieval methods and their characteristics and filters that are applied is given in Tab. 2.

### 3.3 Turbulence estimation from airborne data

The estimation of turbulence parameters from the wind measurement system on the DLR Cessna Grand Caravan 208B (Mallaun
et al., 2015) is done very similarly to the in-situ estimations from the sonic anemometer. TKE is calculated from the sum of variances as described in Sect. 3.3. Dissipation rate is also calculated from the second order structure function, but with different bounds for the time lag. For the flight data we use $\tau_1 = 0.2$ s and $\tau_2 = 2$ s, corresponding to approximately 13-130 m lag at $65\,\mathrm{m\,s^{-1}}$ mean airspeed.

To evaluate the heterogeneity of turbulence due to changing land use along the flight legs of more than 50 km length, we
divided the legs into sub-legs of 6.5 km (i.e. 100 s averaging time) and calculate turbulence for each leg individually. The location of the legs and the sublegs is shown in Fig. 2.





**Table 2.** Overview of turbulence retrieval methods and the applied filters and methods

|  | E89 | S17A | S17 | W19 |
|---|---|---|---|---|
| TKE | yes | yes | yes | yes |
| $\varepsilon$ | no | yes | yes | yes |
| lidar volume averaging effect | no | no | yes | yes |
| CNR filter | yes | yes | yes | no |
| filter of bad estimates | no | no | no | yes |
| integral length scale filter | yes | yes | yes | yes |
| advection filter | no | no | yes | no |
| advection correction | no | no | no | yes |
| variance modifications | yes | yes | yes | yes |

## 4 Validation

### 4.1 Comparison to sonic anemometer

Best possible validation of the methods introduced in Sect. 3 can be performed with the lidar in close proximity to the meteorological mast such as at the measurement site at MOL-RAO. The sonic anemometers at 50 m and 90 m on the mast almost coincide with measurement levels of the lidar at 52 m and 93.6 m respectively. Since the VAD-retrieval with elevation angle of 35.3° yields TKE as well as its dissipation rate, both turbulence parameters can be compared to values obtained from the sonic anemometers. In this section we will evaluate the methods described in Sect. 3.2 and in particular the validity of the assumptions made in Sect. 3.2.1 and the efficiency of the advection correction described in Sect. 3.2.3.

### 4.1.1 Validation of modified variance

In Sect. 3.2.1 we introduced two modifications on the calculation of the averaged lidar radial wind speed variances. These changes are especially necessary if a low number of VAD scans is used for turbulence retrieval. In the MOL-RAO experiment, VAD scans are run continuously with $\varphi = 35.3°$, so that the modifications can be tested against the original version of the retrieval method. The sonic anemometer at 90 m of the meteorological mast serves as an independent validation measurement. Figure 5a shows a time series of the two methods and the sonic anemometer in a time period in which all systems were providing good data almost without interruption (22 April - 29 April 2019). The lidar retrieval with both variance methods follows the sonic anemometer TKE estimation very well through the diurnal cycles, with some occasional overestimation that will be discussed in Sect. 4.1.2. Figure 5b shows the scatterplot which directly compares the S17 retrieval calculated with azimuth averaged variances to the S17 retrieval using total variances. There is a higher estimation of TKE in the total variance method which increases with TKE. We cannot fully explain this effect at this point, but it might be due to the small scale





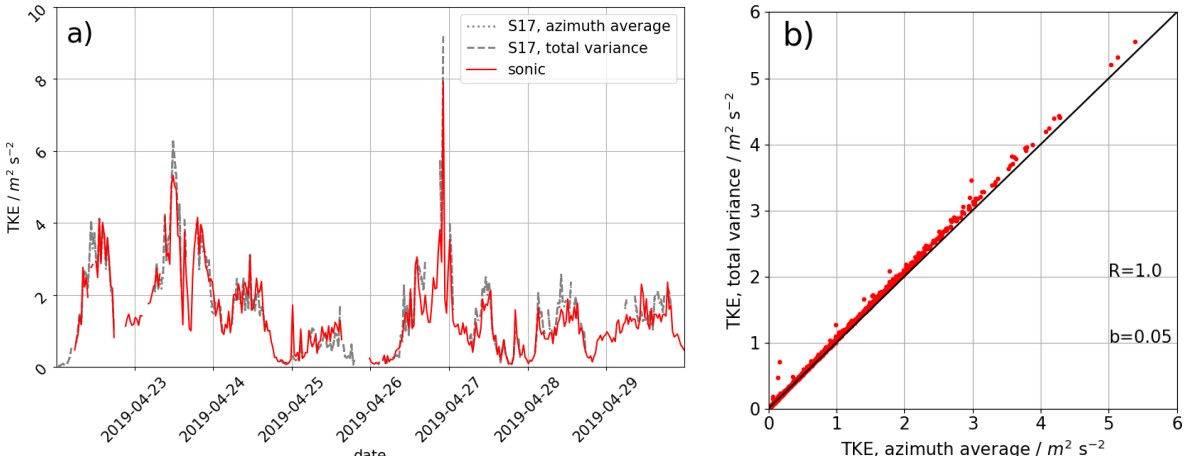

**Figure 5.** Time series of TKE from lidar retrievals compared to a sonic anemometer at 90 m above ground level (a) and scatterplot comparing the lidar retrieval with averaged variances at specific azimuth angles $\theta$ to the modified, total variance method (b).

turbulence that cannot be resolved with the 72 s sampling rate of radial wind speeds at individual azimuth angles. It is small enough to be neglected for further analysis.

### 4.1.2 Comparison of lidar retrievals

The MOL-RAO dataset allows us to compare the retrievals without consideration of lidar volume averaging (E89 and S17A)
to the S17 retrieval and its modified version W19 introduced in this study. For this purpose, the individual retrieval results are compared to the sonic anemometer estimates of TKE and its dissipation rate $\varepsilon$. Figure 6 shows the scatter plots for TKE at the two measurement levels. For each method, the coefficient of determination $R_c^2$ of the linear regression between sonic measurement and lidar retrieval is given, as well as a bias which is calculated as $b = \overline{(y - x)}$ for TKE and $b_{\log} = \overline{(\log_{10} y - \log_{10} x)}$ for $\varepsilon$. We find that with the E89-method, TKE is systematically underestimated, as expected. In contrast to that, the S17-method
yields slightly overestimated TKE-values, if no advection filter is applied (light red dots), but a good agreement with the sonic anemometer in the absence of advection (red dots). The overestimation of TKE is larger for the lower level at 50 m compared to the 90-m level which we attribute to the smaller averaging volume $\Delta y$. Our refined version of the retrieval, including the advection correction improves the results slightly and especially for the high turbulence cases (corresponding to high wind speeds) at the 50-m level, as the scatterplots show. Figure 7 gives the scatterplot comparison of $\varepsilon$-retrieval from S17A, S17
and W19 with the sonic anemometer respectively. E89 does not provide an estimate for $\varepsilon$. Even more clearly than for TKE, the underestimation of the method without consideration of lidar volume averaging (S17A) is found. Also, a now positive bias of $b = 0.09 \ m^2 \ s^{-2}$ of lidar estimates with the S17-method at the 50-m level is reduced with the advection correction to $b = 0.06 \ m^2 \ s^{-2}$. It is evident from the $\varepsilon$-estimates that all lidar retrievals underestimate turbulence significantly compared to



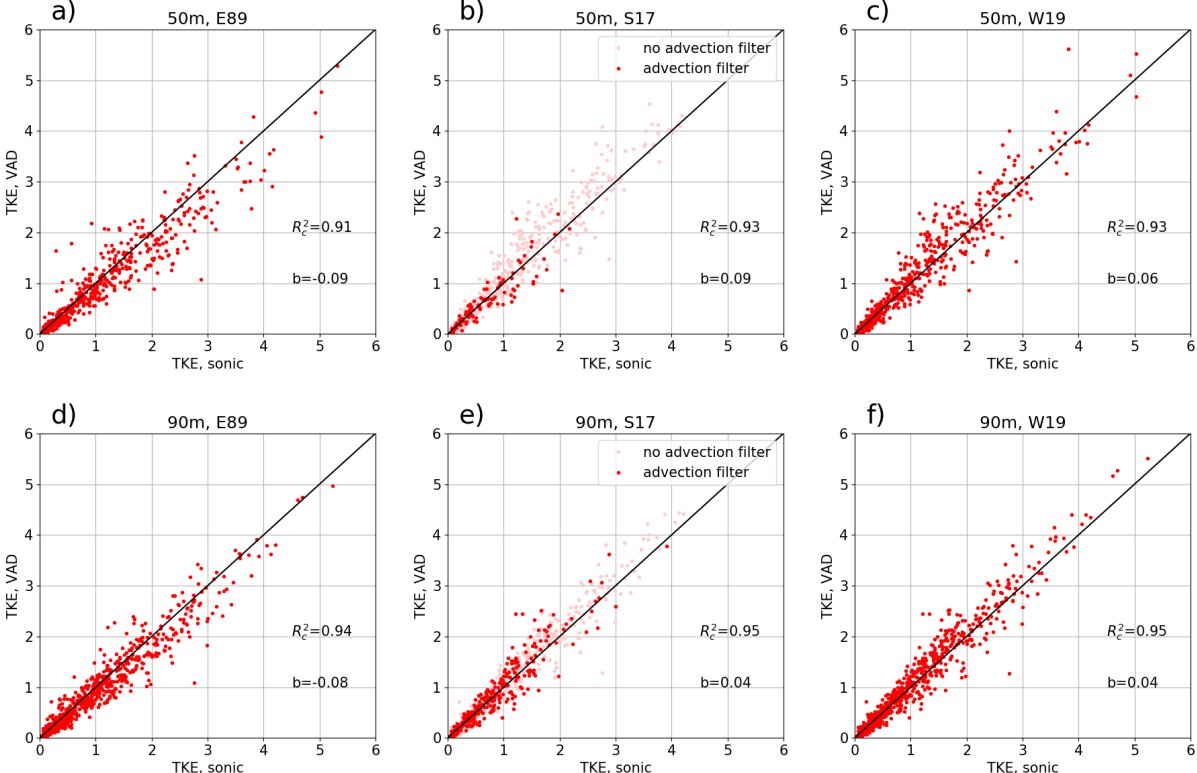

**Figure 6.** Scatter plot of lidar TKE retrieval against sonic anemometer TKE at 50-m level (a-c) and 90-m level (d-f). E89-retrieval is shown in (a) and (d), S17-retrieval in (b) and (e) and W19-retrieval in (c) and (f). The light red dots in (b) and (e) show all TKE estimates without filter for advection, the dark red dots have the advection filter applied.

the sonic anemometer in the low turbulence regimes, and in particular for values smaller than $10^{-3}$ $\mathrm{m^2\,s^{-3}}$, which is why these values have been excluded for the estimation of biases (grey dots).

### 4.1.3 Evaluation of advection error

To evaluate the error that is caused by advection in the S17-retrieval, all data that was collected at MOL-RAO was binned into
5   wind speeds with a bin-width of 1 m s$^{-1}$. The mean absolute error between the lidar retrievals and the sonic anemometers at the respective level is calculated and shown in Figure 8 for TKE and $\varepsilon$. Although the averaged errors of TKE are small in general, it shows that the W19-method does reduce the error in comparison to the S17 method at the 50-m level. The error for $\varepsilon$ at the 50-m level increases with wind speed, but less for the W19-method. Hardly any improvement is found for the already small errors of TKE and $\varepsilon$ at 90 m.





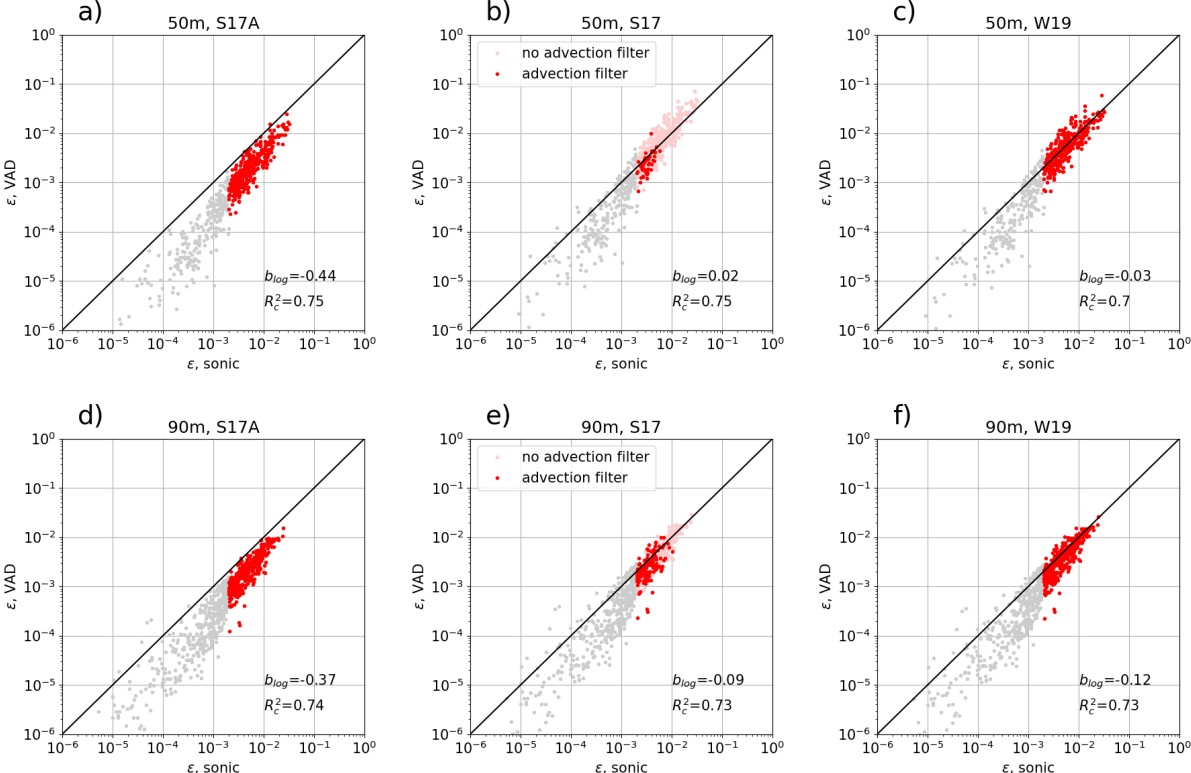

**Figure 7.** Scatter plot of lidar dissipation rate retrieval against sonic anemometer dissipation rate. The light red dots in (b) and (e) show the results without advection filter. The light grey dots are all estimates below $2 \cdot 10^{-3}$ m$^2$s$^{-3}$. Grey dots are not used for calculation of $R$ nd $b_{\log}$.

## 4.2 Comparison of elevation angles

VAD-scans with $35.3°$ allow to retrieve TKE as well as momentum fluxes using the methods described in Sect. 3. The disadvantage compared to VAD scans with larger elevation angles is that at the same range of the lidar line-of-sight measurement, lower altitudes are reached. If the limit of range is not given by the ABL height in any case, this can lead to significantly lower

5 data availability at the ABL top. Another advantage of greater elevation angles is that the horizontal area that is covered with the VAD and thus the footprint of the measurement is much smaller than with low elevation angles. From the theory derived in Sect. 3.3, we see that the dissipation rate retrieval does not depend on the elevation angle and can thus be obtained from VAD scans with $75°$ elevation angle as well if the assumptions of isotropy and homogeneity hold. However, since the VAD at $75°$ has the more narrow cone, the separation distances $\Delta y$ at respective measurement heights are smaller and thus the sensitivity

10 to advection errors is expected to be larger. The measurements with two different elevation angles at the CoMet-campaign allow to compare dissipation rate retrievals for both kinds of VAD scans with the restriction that they are not simultaneous, but





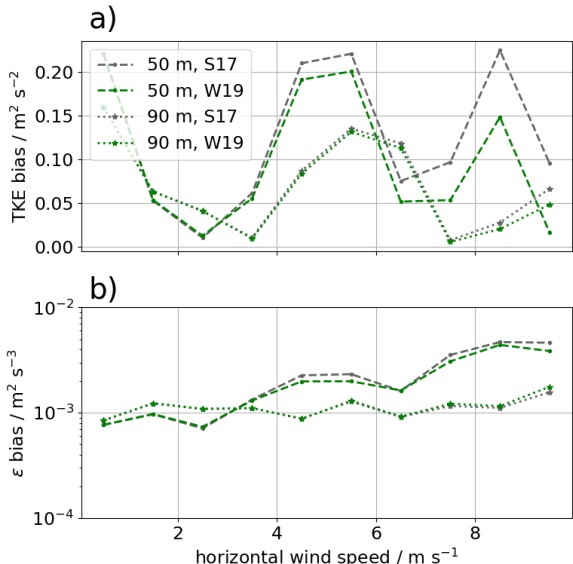

**Figure 8.** Difference of lidar retrieval of TKE (a) and TKE dissipation rate (b) compared to sonic anemometer as a function of wind speed.

sequential.

Figure 9 shows the comparison of both types of VADs in scatter plots of three measurement heights for the whole campaign period and DLR#1. In general, a large scatter is found between the two types of VAD which can be attributed to the different measurement times, different footprint and heterogeneous terrain. Applying a filter for significant advection as described in

Sect. 3.2.4 removes most of the measurement points at the 100 m level in the 75° VAD. Without the filter (grey and red points), large, systematic overestimation against the VAD at 35.3° is found, which can still be seen at 500 m, but is not found at 1000 m any more. With the advection correction of W19, the systematic error is reduced, but the random errors remain.

As for the comparison with the sonic anemometer, we evaluate the error in dependency of wind speed by binning the data in wind speeds between $0\,\mathrm{m\,s^{-1}}$ and $10\,\mathrm{m\,s^{-1}}$ (see Fig. 10). A clear trend is found for the 100 m-level which can be significantly

reduced with the W19 method. A very small difference between the two elevation angles at the 500-m level is only reduced very little in the W19 method.

## 4.3    Comparison to D-FDLR

At CoMet, no meteorological tower with sonic anemometers on levels that could be compared to the Doppler lidars was available. Instead, the aircraft D-FDLR was operating with a turbulence probe and provided in-situ turbulence data. On 5 June

2018, the aircraft was flying a so-called 'wall'-pattern, with long, straight and level legs at five altitudes (800 m, 1000 m, 1100 m, 1300 m and 1600 m). At least the lowest three levels of this flight allow a comparison to measurement levels of the top levels of lidar measurements on this day. Figure 11 shows the measured TKE of D-FDLR in the five flight levels. Only at

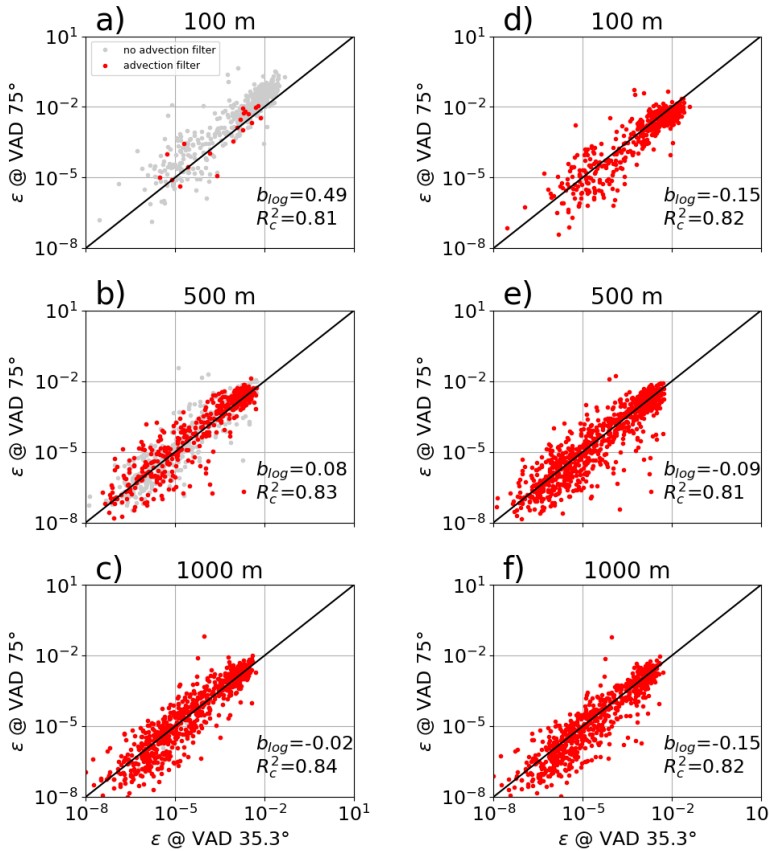

**Figure 9.** Scatter plot of lidar dissipation rate retrieval from VAD scans with $75°$ elevation angle versus $35.3°$. On the left, retrievals without advection correction are shown for the three different levels (a) 100m, (b) 500m and (c) 1000m. On the right the corresponding scatter plots with advection correction are presented (d-f).

the lowest two light levels (i.e. 800 m and 1000 m), significant turbulence is measured, with strong variations along the flight path.

Figure 12 shows the measurements from D-FDLR and the lidars DLR#1 and DLR#3 that were taken between 1400 and 1530 UTC as vertical profiles. The solid red line gives the average of all sub-legs along the 50 km flight of D-FDLR, and the shaded areas give the range between the minimum and the maximum at each height. It shows nicely how the measurements of turbulence at the DLR#3-site are significantly lower than at DLR#1, which we attribute to the lake fetch. The D-FDLR measurements of TKE almost match with the DLR#1 site and are all higher than at the DLR#3-site, which fits to the environmental conditions of heterogeneous land-use. Figure 12a also gives the comparison between E89-, S17- and W19-estimates of the same dataset. Here, it shows that the difference between S17 and W19 only occurs at the very lowest level, but the underestimation of the E89-method is found up to 750 m. In dissipation rate estimates, the DLR#1 measurements are at the





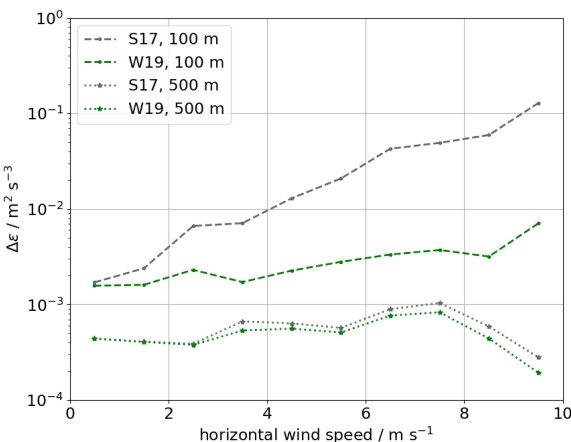

**Figure 10.** Difference of lidar $\varepsilon$-retrievals of both kinds of VAD as a function of wind speed.

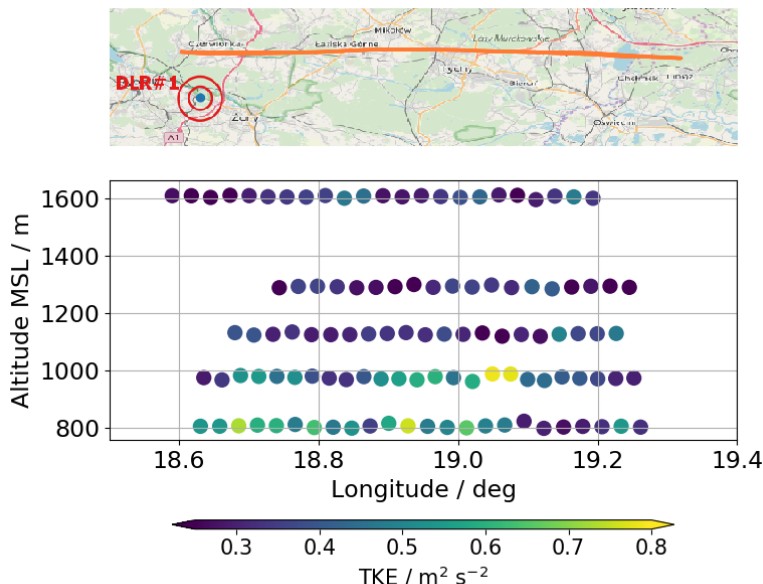

**Figure 11.** D-FDLR TKE measurements at five flight levels. Map data ©OpenStreetMap contributors 2019. Distributed under a Creative Commons BY-SA License.





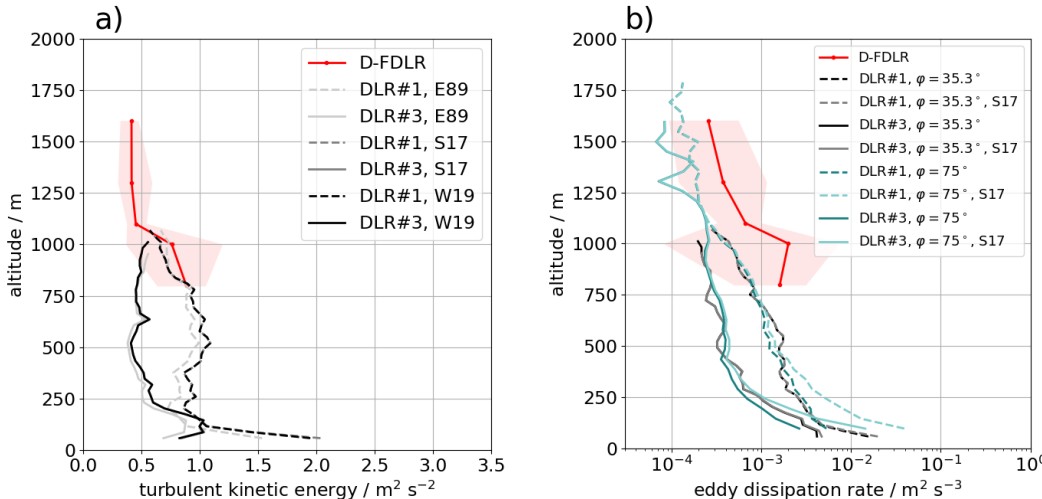

**Figure 12.** Vertical profile of TKE (left) and $\varepsilon$ (right) compared to measurements by D-FDLR.

low end of the range that was measured with D-FDLR and the lake-site measurements are even smaller. The estimates from $35.3°$-scans and $75°$-scans agree very well, especially at the higher levels, which shows that the assumption of isotropy and homogeneity seem to hold. The presumed underestimation of $\varepsilon$ of lidar retrievals compared to in-situ measurements at absolute values of $10^{-3}\,\mathrm{m^2\,s^{-3}}$ is consistent to what was found for the comparison to sonic anemometer measurements. This single case

of airborne measurements compared to lidar retrievals at higher altitudes can however not provide any statistical validation.

Figure 13 shows measurements of $\varepsilon$ retrieved with the W19-method for the VAD scans with $75°$ elevation and all three lidars. It shows that the growth of the boundary-layer with its increased turbulence can be nicely captured by the lidars. There are some differences between the three locations, especially lower turbulence close to the ground at the DLR#3-location and a higher boundary layer at the DLR#2 location. More studies will be necessary in future, analyzing the data of the whole

campaign to improve the understanding of land-atmosphere interaction in this case.

## 5  Conclusions

In this study, we used four different methods to retrieve turbulence from the same data obtained through lidar VAD scans. The MOL-RAO experiment allowed us to show that methods which do not account for the lidar volume averaging effect underestimate turbulence compared to sonic anemometers at 50 m and 90 m systematically. The S17-method tackles this

problem, but introduces an overestimation in our dataset. Parts of this overestimation can be attributed to advection, which distorts the retrieval of the azimuth structure function and the transverse filter function in the lidar model. The advection effects are most relevant at the lowest measurement heights where the spatial separation of lidar beams along the VAD cone $\Delta y$ are small. We propose a correction for this issue and show here that our method reduces the systematic errors compared to the





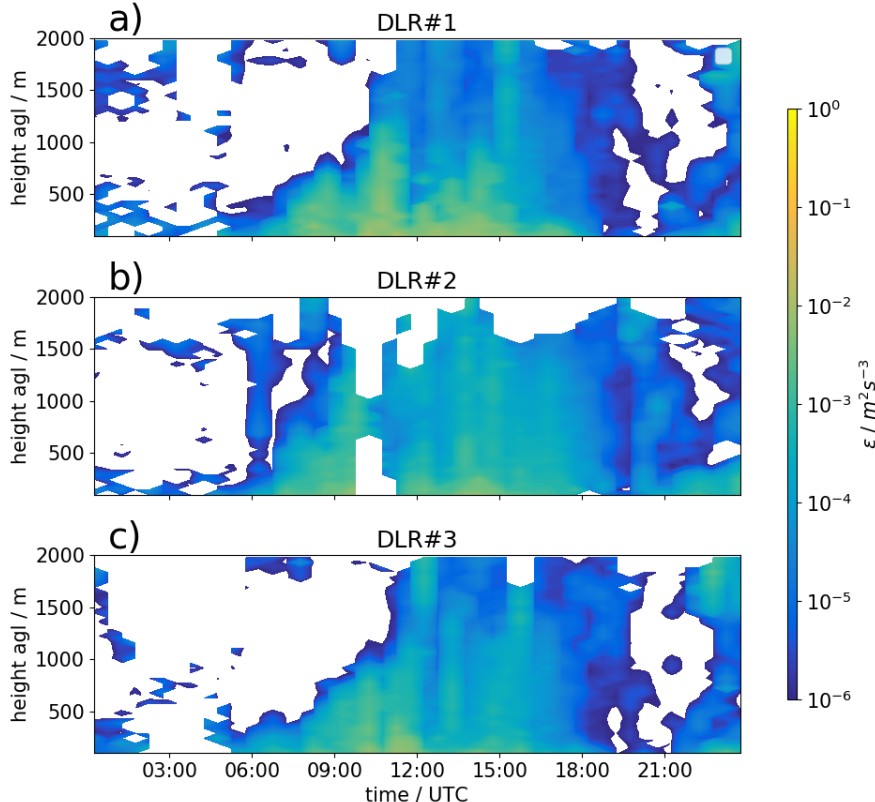

**Figure 13.** Diurnal cycle of TKE dissipation rate on 5 June 2018 at the three lidar locations calculated with the W19-method.

sonic anemometers at the 50-m level.It is also shown that the bias increases with wind speed. With all retrievals, dissipation rates of values smaller than $10^{-3}$ $\mathrm{m^2\,s^{-3}}$ are underestimated by the lidars, likely because the small scale fluctuations that are carrying much of the energy in these cases, cannot be resolved any more. A remaining piece of uncertainty are the lidar parameters $\Delta p$ and $T_w$ which are given by the manufacturer for the lidar type, but could potentially differ for individual lidars.

5 Exact knowledge about these parameters could reduce the uncertainty of the model functions $F$ and $A$ (see App. B) and thus improve the corrections of the volume averaging effects. It is conceivable that the observed overestimation of the S17 (and W19) based TKE can partly also be attributed to these uncertainties.

The aircraft measurements that were carried out during the CoMet campaign were used to show the agreement of the lidar retrievals with in-situ measurements at higher altitudes. Unfortunately, only measurements of one day allowed a comparison

10 and the spatial separation of the measurements introduces additional uncertainty. It was found that TKE estimates from lidar and aircraft compare rather well, but the small values of dissipation rates at these heights are underestimated by the lidar to a similar order of magnitude as for low turbulence conditions in the sonic anemometer comparison. Dedicated experiments will be necessary in future to provide more comprehensive validation datasets for turbulence retrievals with lidar VAD scans. Given





the larger separation distances $\Delta y$ of the lidar beams at higher altitudes, the assumption that $l\Delta y \ll L_v$ is more likely to be violated. Airborne in-situ measurements are the best way to validate the assumptions and the lidar retrievals in these cases.

The CoMet dataset was also used to show that with VAD-scans with larger elevation angle (here: 75°) can be used to retrieve TKE dissipation rate with the same method as for VAD-scans with 35.3° and the results are comparable. For this narrow

5  VAD cone scans, we showed that the advection correction is much more important than for lower elevation angles and strong overestimation of $\varepsilon$ can occur in conditions with high wind speeds if it is not applied. The distribution of three lidars in Upper Silesia in areas of different land-use shows the variability of turbulence and boundary-layer flow in this area. Using the VAD scans with different elevation angle can in future potentially help to analyze horizontal heterogeneity in the boundary layer and its impact on the calculation of area-averaged fluxes.

10  *Data availability.* The data are available from the author upon request.

## Appendix A:  Fourier decomposition

Eberhard et al. (1989) show that radial wind speed variance can be decomposed into the components $u$, $v$ and $w$ of the meteorological wind vector:

$$
\begin{aligned}
\langle v_r^2(R,\theta,\varphi)\rangle &= \langle [V_r(R,\theta,\varphi,t) - \langle V_r(R,\theta,\varphi)\rangle]^2\rangle \\
&= \frac{\cos^2\varphi}{2}\left[\langle u^2\rangle + \langle v^2\rangle + 2\tan^2\varphi\langle w^2\rangle\right] \\
&\quad + \sin 2\varphi\langle uw\rangle - \sin 2\varphi\langle vw\rangle\sin\theta \\
&\quad + \frac{\cos^2\varphi}{2}\left[\langle u^2\rangle + \langle v^2\rangle\right]\cos 2\theta - \cos^2\varphi\langle uv\rangle\sin 2\theta
\end{aligned}
\tag{A1}
$$

A partial decomposition of Eq. A1 yields:

$$
\begin{aligned}
\langle u^2\rangle + \langle v^2\rangle + 2\tan^2\varphi\langle w^2\rangle &= \frac{1}{\pi\cos^2\varphi}\int_0^{2\pi}\langle v_r^2\rangle d\theta \\
&= \frac{2}{\cos^2\varphi}\langle\langle v_r^2\rangle\rangle_\theta
\end{aligned}
\tag{A2}
$$

$$
\langle uw\rangle = \frac{1}{\pi\sin^2\varphi}\int_0^{2\pi}\langle v_r^2\rangle\cos\theta d\theta
\tag{A3}
$$

$$
\langle vw\rangle = \frac{-1}{\pi\sin^2\varphi}\int_0^{2\pi}\langle v_r^2\rangle\sin\theta d\theta \quad .
\tag{A4}
$$

These equations provide the basis for the retrieval of TKE and momentum fluxes from lidar VAD measurements.





## Appendix B: Lidar filter functions

Theoretical models for the spectral broadening of lidar measurements ($F(\Delta y)$) and the structure function($A(l\Delta y)$) are derived in Banakh and Smalikho (2013) from the two-dimensional Kolmogorov spectrum for lidar measurements of turbulence:

$$\Theta(\kappa_z, \kappa_y) = C_3(\kappa_z^2 + \kappa_y^2)^{-4/3}\left[1 + \frac{8}{3} \cdot \frac{\kappa_y^2}{\kappa_z^2 + \kappa_y^2}\right] \tag{B1}$$

$$F(\Delta y) = \int\limits_0^\infty d\kappa_z \int\limits_0^\infty d\kappa_y \Theta(\kappa_z, \kappa_y)\left[1 - H_\parallel(\kappa_z)H_\perp(\kappa_y)\right] \tag{B2}$$

$$A(l\Delta y) = 2\int\limits_0^\infty d\kappa_z \int\limits_0^\infty d\kappa_y \Theta(\kappa_z, \kappa_y)H_\parallel(\kappa_z)H_\perp(\kappa_y)\left[1 - \cos(2\pi l\Delta y_i \kappa_y)\right] \quad, \tag{B3}$$

where $H_\parallel$ is the longitudinal and $H_\perp$ the transverse filter function of lidar measurements in the VAD scan:

$$H_\parallel(\kappa_1) = \left[\exp\left[-(\pi\Delta p\kappa_1)^2\right]\frac{\sin(\pi\Delta R\kappa_1)}{\pi\Delta R\kappa_1}\right]^2 \tag{B4}$$

$$H_\perp(\kappa_2) = \left[\frac{\sin(\pi\Delta y\kappa_2)}{\pi\Delta y\kappa_2}\right]^2 \quad, \tag{B5}$$

10    where $\Delta p$ is derived from the FWHM pulse width $\tau_p$, $\Delta R$ from the time window $T_w$ and $\Delta y$ from the VAD azimuth increment $\Delta\theta$:

$$\Delta p = 0.5c\left(\frac{\tau_p}{2\sqrt{\log 2}}\right) \tag{B6}$$

$$\Delta R = 0.5cT_w \tag{B7}$$

$$\Delta y = R\Delta\theta\cos\varphi \tag{B8}$$

$$\tag{B9}$$

## Appendix C: Sum of variance of subsamples

For statistically independent subsamples $X_j$ with size $k_j$, the total variance of the dataset can be derived as follows:

$$E_j = \mathrm{E}\left[X_j\right] = \frac{1}{k_j}\sum_{i=1}^{k_j} X_{ji} \tag{C1}$$

$$V_j = \mathrm{Var}\left[X_j\right] = \frac{1}{k_j - 1}\sum_{i=1}^{k_j}(X_{ji} - E_j)^2 \tag{C2}$$





$$\mathrm{Var}\left[X\right] = \frac{1}{n-1}\sum_{j=1}^{g}\sum_{i=1}^{k_j}(X_{ji} - \mathrm{E}\left[X\right])^2 \tag{C3}$$

$$= \frac{1}{n-1}\sum_{j=1}^{g}\sum_{i=1}^{k_j}\left((X_{ji} - E_j) - (\mathrm{E}\left[X\right] - E_j)\right)^2 \tag{C4}$$

$$= \frac{1}{n-1}\sum_{j=1}^{g}\sum_{i=1}^{k_j}(X_{ji} - E_j)^2 - 2(X_{ji} - E_j)(\mathrm{E}\left[X\right] - E_j) + (\mathrm{E}\left[X\right] - E_j)^2 \tag{C5}$$

$$= \frac{1}{n-1}\sum_{j=1}^{g}(k_j - 1)V_j + k_j(\mathrm{E}\left[X\right] - E_j)^2 \quad , \tag{C6}$$

where $n = \sum k_j$. Eventually, for equally sized subsamples one obtains:

$$\mathrm{Var}\left[X\right] = \frac{1}{n-1}\sum_{j=1}^{g}(k-1)V_j + k(g-1)\mathrm{Var}\left[E_j\right] \tag{C7}$$

$$= \frac{k-1}{n-1}\sum_{j=1}^{g}V_j + \frac{k(g-1)}{k-1}\mathrm{Var}\left[E_j\right] \tag{C8}$$

## Appendix D: Validation of wind retrieval

The FSWF-retrieval is used to obtain the three-dimensional wind vector from the lidar VAD scans. The results of the retrieval is compared to the sonic anemometers at 50 m and 90 m and shown in Fig. D1. To show the distortion of the mast, no data have been removed in the retrieval of wind speed and wind direction for this figure.





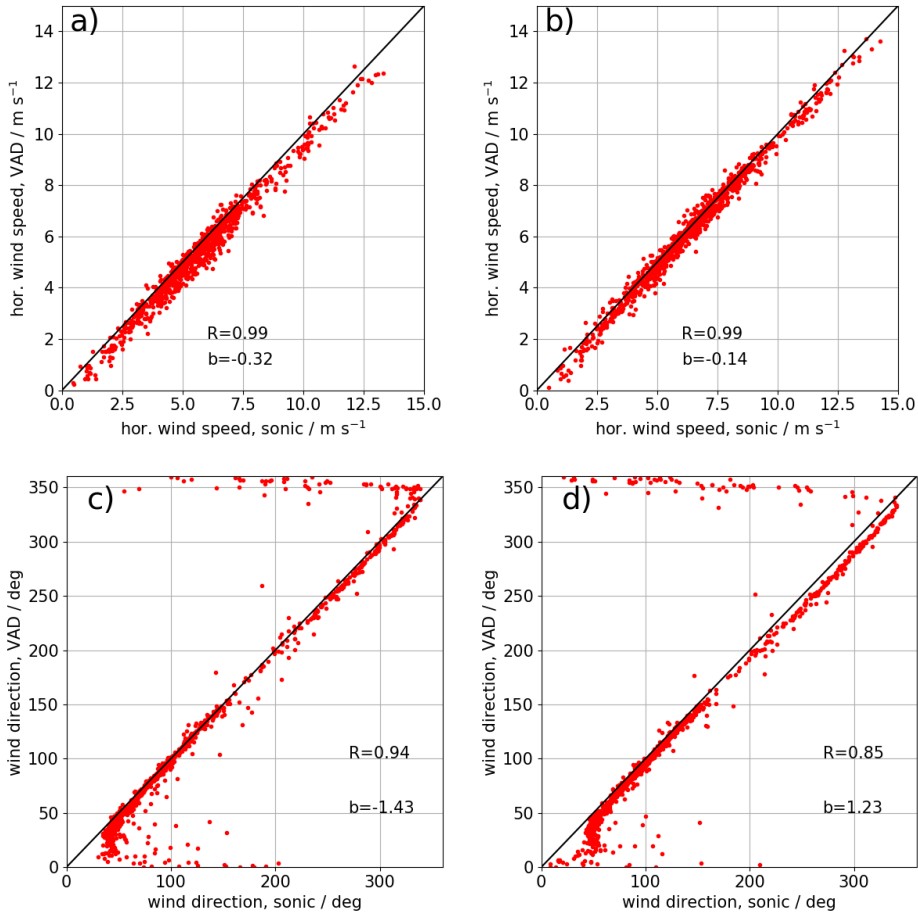

**Figure D1.** Scatter plot of horizontal wind speed retrieved from lidar measurements compared to sonic anemometer measurements at 50 m (a) and 90 m (b) and wind direction (c and d).





## Appendix E: Nomenclature

| | |
|---|---|
| $\lambda$ | laser wavelength |
| $\tau_p$ | lidar pulse length (full width half maximum, FWHM) |
| $T_w$ | lidar time window |
| $\varphi$ | elevation angle |
| $\theta$ | azimuth angle |
| $u$ | wind component towards East |
| $v$ | wind component towards North |
| $w$ | upward wind component |
| $\sigma_u^2$ | $u$-wind component variance |
| $\sigma_v^2$ | $v$-wind component variance |
| $\sigma_w^2$ | $w$-wind component variance |
| $\tau_i$ | time separation |
| $V_r$ | radial wind component |
| $v_r$ | radial wind component difference from mean |
| $\sigma_r^2$ | radial wind component variance |
| $R$ | range gate distance |
| $E_{\text{TKE}}$ | turbulence kinetic energy |
| $\varepsilon$ | TKE dissipation rate |
| $C_K$ | Kolmogorov constant |
| $\psi$ | azimuth angle increment |
| $\kappa$ | wave number |
| $r$ | separation distance |
| $\sigma_L^2$ | variance of lidar measurements |
| $\sigma_e^2$ | lidar instrumental noise |
| $\sigma_a^2$ | variance of lidar measurements without instrumental noise |
| $\sigma_t^2$ | turbulent broadening of the Doppler spectrum |
| $\Delta R$ | distance between neighboring range gate centers |
| $D_r$ | azimuth structure function |
| $D_s$ | longitudinal structure function measured by sonic anemometer |
| $D_L$ | lidar measurement of azimuth structure function |
| $D_a$ | $D_L - 2\sigma_e^2$ |
| $A$ | theoretical model for azimuth structure function |
| $F$ | theoretical model for turbulent broadening of the Doppler spectrum |
| $\Delta y$ | distance of lidar beam movement during one accumulation period |
| $\Delta y_c$ | modified $\Delta y$ for advection |
| $p_M$ | model PDF |
| $P$ | probability of bad estimates |
| $L_v$ | integral length scale |
| $\omega_s$ | angular velocity of VAD scan |
| $\Psi$ | wind direction |
| $U$ | horizontal wind speed |
| $\Delta t$ | accumulation time |
| $R_c$ | linear regression correlation coefficient |
| $b$ | measurement bias |



*Author contributions.* NW, EP, AR and CM helped design and carry out the field measurements. CM provided processed data from the D-FDLR aircraft turbulence probe. NW analyzed the data from the sonic anemometers, the profiling lidars and the D-FDLR. NW wrote the paper, with significant contributions from EP. All the coauthors contributed to refining the paper text.

*Competing interests.* The authors declare that they have no competing interests.

5   *Acknowledgements.* We want to thank Jarosław Nęcki and all of the students of the University of Science and Technology, Cracow for their tireless work in support of the CoMet campaign. We thank the Aeroklub Rybnickiego Okręgu Węglowego, Hotel Restauracja Pustelnik and Agroturystyka "Na Polanie" for providing space and infrastructure for the lidar deployment in Upper Silesia. We thank Frank Beyrich and Andreas Fix for internal reviews and their input to the manuscript.



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
