# Peer review of "Towards improved turbulence estimation with Doppler wind lidar VAD scans"

_Atmospheric Measurement Techniques, 2020_

## Referee Comment (RC1) · Anonymous Referee #1 · 1 Apr 2020

This paper discusses methods for retrieval of turbulence kinetic energy and dissipation rate from VAD (i.e. conical) Doppler lidar scans. Starting from established methods (structure function methods), they introduce changes that permit retrievals using a smaller number of scans, and a correction for the effects of advection. The methods are evaluated by comparison sonic measurements in one field campaign.

Through the comparison with sonics they are able to show a slight improvement when pulse averaging effects are considered (Fig 6). The main novelty here seems to be the advection correction, which they claim improves the retrievals, but the results are a bit underwhelming (see Fig. 6 and 7). I'm not at all convinced that the tiny improvements in the metrics (bias and correlation) are significant. I would be more inclined to conclude that the advection correction doesn't have a significant effect.

The organization of the paper is fine, and it is written well enough to be understood. However, I do feel that the authors could have done a better job at explaining a number of things, which I highlight below. As it stands the paper requires fair significant revision before acceptance.

Abstract: The first 3 or 4 sentences could be probably be reduced to a single sentence in favor of allowing for a more quantitative summary later in the abstract. As it stands, the abstract lacks sufficient substance. The author should incorporate more hard results from the comparisons with sonic anemometers.

page 4 line 3: Not everyone will know where Upper Silesia is (including myself until I looked it up), I suggest "…were installed in Upper Silesia, in southern Poland (or where ever), …"

page 4 line 15: change "…and were finally fixed…" to "…and were finally choosen…"

Table 1. The Stream Line wavelength is 1.548 μm.

Section 3.2: The author should include the equation for the measured azimuth structure function – since this is key for the dissipation rate retrieval methods.

Equation 3: The condition that $\varphi = 35.3°$ should be made more explicit to prevent possible misused. I suggest something like

$$E_{TKE} = \frac{3}{2} \bar{\sigma}_r^2 \Big|_{\varphi=35.3^o}$$

Page 6, lines 4-6: The author states that the "TKE dissipation rate is estimated through a fit of the measured second-order structure function of horizontal velocity to the theoretical …" This statement implies that the observations are adjusted to fit the model, when in fact it's the other way around, i.e. the model parameters are adjusted in order to fit the observations. Please rephrase.

Page 6, lines 23-24: Similar to last comment. The author states that "A fit of the azimuth structure to the equation …" again implies that the observations are being adjusted to fit the model, when in fact it's the other way around. Please rephrase.

Page7, line 1: The author states that "Scanning with Doppler lidar in a VAD implies a volume averaging of radial velocities in longitudinal and transverse directions." Aside from the grammatical errors, this statement is not generally true because transverse averaging is not an issue for step-stair scans, only for continuous motion scans. The author should briefly mention the two different types of scans in their introduction. Also, the author should define what they mean by longitudinal and transverse (i.e. along the beam, and orthogonal to the beam).

Page 7 line 5: change "… radial wind speed…" to " …radial velocity…". Wind speed is a (positive) scalar, velocity is a vector. In this sentence your talking about the radial component of the velocity vector. "Radial wind speed" makes no sense.

Page 7 starting at line 7: The discussion here is a bit disjointed and difficult to follow. Equations 5-7 should be listed after the sentence on line 5 (starting with "It is based on the decomposition…"). As it is, these equations are introduced without any corresponding text. One suggestion might be …

"In Smalikho and Banakh (2017), this method has been combined with the E89-method to yield TKE, and the momentum fluxes. It is based on the decomposition of radial velocity variance into its subcomponents, i.e.

$$\sigma_L^2 = \sigma_a^2 + \sigma_e^2,$$

$$\sigma_a^2 = \sigma_r^2 - \sigma_t^2,$$

and

$$\sigma_r^2 = \sigma_L^2 + \sigma_t^2 - \sigma_e^2$$

where $\sigma_L^2$ is the lidar measured variance, $\sigma_a^2$ is the lidar measured variance without instrumental error, $\sigma_e^2$ is the instrumental error variance, and $\sigma_t^2$ is turbulent broadening of the lidar measurement. In Smalikho and Banakh (2017), all of these variances and corresponding structure functions are calculated for single azimuth angles and then averaged."

Page 7, line 17: Recommend changing "Substituting $\sigma_e^2$ in Eq. 7 with Eq. 8 yields:" to "Combining Eq. 7 with Eq. 8 yields:"

Page 7, lines 18-23, including equations 9, 10 and 11: There is a dependence on the separation distance on the right side of equation 9 that presumably cancels such that the right side is effectively constant, i.e. independent of separation distance. This is a subtle point that is not made by the author. Also, in equation 10, the author has substituted $\psi_l$ with $\psi_1$ without any explanation or justification. Please explain.

Page 8, line 10-11: The author states "…from VAD scans with other elevation angles as well." You should be a bit more specific here, since readers may not know what you mean by "other elevation angles." I assume you're referring to elevation angles different from 35.3º.

Page 8, line 13-15: The author states "The value of l = 9 is chosen following the example of Smalikho and Banakh (2017) and corresponds to $l\Delta\theta = 9^o$ as it was found to be suitable in all conditions in that study." The discussion up to this point had been fairly general. Now, suddenly the author is referring to a very specific VAD scan. The author should be a bit more specific as to which scan (and which experiment) they are referring to.

Page 8, line 24: change "…radial wind speeds…" to "…radial velocities…"

Page 8, line 28: change "…radial wind speeds…" to "…radial velocities…"

Page 10, line 1: change "…radial wind speeds…" to "…radial velocities…"

Page 10, line 3: change "…radial wind speed…" to "…radial velocity…"

Page 10, line 6: Recommend changing "Since the mean of the radial wind speed fluctuations $v_f$ = 0 by definition, it is:" to something like "Since the mean of the radial velocity fluctuations is zero by definition, equation (20) becomes "

Page 10 line 5: The author states "(here: g=360 for all azimuth angles)". The reference to a specific value of g here is a bit perplexing. Please explain.

Equation 20: The summation is over j, but there is no dependence on j in the quantity being summed. Please explain.

Page 10, lines 11-12: change "…radial wind speeds…" to "…radial velocities…"

Page 10, line 16-17: The author states "Measured PDFs of the variables … are fit to the model PDFs to obtain an estimation of the corresponding standard deviations $\sigma_1$, $\sigma_2$ and $\sigma_3$ and probability of bad estimates $P_1$, $P_2$ and $P_3$." This statement implies that the observations are fit to the model. In other words, the observations are tweaked to get agreement with the model. That's certainly not what is happening. Please rephrase.

Section 3.2.2: It seems to me there is some slightly circular logic going on here. From what I gather, your fitting equation 22 to the measured PDFs to obtain estimates of the variance and the false-alarm probability. But since the real distributions aren't Gaussian, you end up computing the variance directly from the data. This begs the question as to why the variance was treated as an adjustable parameter in the first place. Why not just compute the variance from the data to begin with and then use equation 22 to estimate only the false alarm probability. What do the distributions look like? How good (or bad) are the fits?

Section 3.2.3: In this section the author throws down a series of equations without adequate discussion. The authors need to do a better job explaining their line of reasoning.

Recommend something like …

"When advection is not considered, the spacing between samples is given by

*equation (25).*

An estimate of the mean spacing can be obtained from

*equation (26),*

where

*equation (27).*

We propose a simplified correction in which

*equation (30),*

where

*equation (28),*

and

*equation (29).*

Appendix C, lines 17: The author introduces the quantity $X_j$ (i.e. a 1D vector), and then in equation C1 it is indicated to be a 2D matrix, i.e. $X_{ij}$. Please explain.

Appendix D: I find no mention of the "FSWF-retrieval" in the paper (did I miss it?). Please elaborate and provide relevant citations.

---

## Referee Comment (RC2) · Anonymous Referee #2 · 16 Apr 2020

"Towards improved turbulence estimation with Doppler wind lidar VAD scans" by Wildmann et al. provides an improved method to retrieve turbulent kinetic energy (TKE) and its dissipation rate using Doppler wind lidar VAD scans. The advantage of the new method is that it corrects also for the effects of advection, which is not included in the previous methods. This feature makes it possible to retrieve turbulent parameters also using higher elevation angles for the VAD scans, here 75º instead of the more commonly used 35.5º. The paper represents clear and thorough work to advance the measurement techniques using Doppler wind lidars and therefore is suitable for publication in AMT. However, before publication it requires major revision mainly due to one aspect in the methodology which has not been sufficiently discussed and which may have a substantial effect on the results.

Major comment:

The method to retrieve $\varepsilon$ from sonic anemometer (and research aircraft) measurements is based on averaging 2 min estimates of $\varepsilon$ over 30 min sampling time. Taking the average means that the distribution of $\varepsilon_{2min}$ is assumed to be Gaussian, which is not necessarily true as the magnitude of $\varepsilon_{2min}$ may vary over several scales of magnitude. When the distribution is not Gaussian, the average will introduce a bias to the $\varepsilon_{30min}$ estimate. Therefore, authors should check the shape of the distribution of $\varepsilon_{2min}$ values during each 30 min period and choose an estimate for $\varepsilon_{30min}$ that is more representative for the distribution of $\varepsilon_{2min}$ values, such as the median of these values.

Minor comments:

Abstract:
It is not obvious to all readers that "DLR Cessna Grand Caravan 208B" is a research aircraft, please, add this to the abstract. Further, it is not clear why do you introduce the research aircraft data in this study as the title is about improvement of turbulence estimation using Doppler wind lidars. For validation purposes research aircraft data cannot be considered as robust as sonic anemometer data. In fact, the explanation for using aircraft measurements is provided only in Section 5 on lines 8-9 of page 21. This explanation should be given already in the abstract but also in the Introduction (Section 1) and maybe also in Section 3.3.

Section 2, page 2, lines 33-34: "*data from two different sites and sets of instruments*":
This is not clear: you use data from four DWLs (three in Upper Silesia, one at Falkenberg), from two sonic anemometers and from one research aircraft. Although you introduce the measurements in two Figures (Figures 1 and 2), it does not change the fact that you have several type of instruments and sites, not just two of each.

Section 2.1., page 3, line 19: remove "*and infrared gas analysers LI7500 (LiCor Inc.)*" because you do not use this data.

Section 2.2, the first two paragraphs on page 4: the description of the CoMet mission is too detailed and not relevant to the topic of this paper, as here the aim is not to investigate $CO_2$ or methane but to develop DWL data processing methods. Please, provide here only such information that is relevant for the present study.

Page 4, line 19: angle should read 35.5º not 35º.

Page 4, lines 20-21, 2 comments:
1. Check the tense of verbs to be consistent.
2. The acronyms of the Doppler lidars in Upper Silesia region are misleading: for the research aircraft you use the acronym "*DLR*" (e.g. in the abstract but also on line 8 of page 4) and for

Doppler lidars you have introduced the acronym "*DWL*". Why do you introduce here another acronym for DWL, i.e., "*DLR*"? Please, use only one acronym for Doppler wind lidars throughout the paper.

Page4, lines 23-25: "*a case study on 5 June 2017, on which D-FDLR was performing long straight and level legs between 800 m and 1600 m as indicated in the flight path in Fig. 2.*" This is the first time you introduce the research aircraft data and it is somewhat vague. Please, provide more information on why did you choose this data set, how did you select this specific period, what kind of instrumentation there was onboard, how accurate are the turbulence measurements from the aircraft in general, etc?

Page 6, line 7: "*the values are calculated for 2-minute intervals and then averaged over half-hour periods.*". Include here information about the distribution of values calculated for 2-minute intervals, to show that the average is (or is not) a representative parameter for the population of values (see also the major comment).

Page 7, line 19: Should it read Eq.2 instead of Eq.3?

Page 7, line 20: Why does $\psi_l$ change to $\psi_1$ here? Please, explain what it means that l=1?

Page 7, line 25: Typo: "*Kolmogorov-Obhukov spectrum*"

Page 8, Equation (15): It is not clear how this Equation is derived from Eq. 13 and 8: what happens to $\sigma_e^2$ that was in Equation 8?

Page 9, line 8: "*In (Smalikho and Banakh, 2017)*" change to "*In Smalikho and Banakh (2017)*"

Page 10, Equation 20: There is no index j in the equation (which is included in the summation). Moreover, are there some parenthesis missing?

Page 12, Section 3.3: Again, it is not clear why research aircraft data is used: is it used a) because it gives more reliable results than DWLs and therefore can be used for validation of DWL data or b) is it used because it would be interesting to know how good the research aircraft data is compared to DWLs (and sonics)?

Page 14, Figure 5: It is not possible to see the dotted line in panel (a). Moreover, in the caption, could you provide the Equation numbers for the averaged variance and total variance methods in order to strengthen the link between the theory and the results.

Page 14, line 5: "*modified version W19 introduced in this study*", maybe you should use acronym W20 for the method introduced in this paper?

Page 15, Figure 6: Does the biases in (b) and (e) include all points or only those after the advection filter?

Page 17, Figure 8: Why there is an oscillatory pattern in TKE bias as a function of horizontal wind speed? The oscillatory pattern is more significant than the differences between the methods, and therefore it should be discussed. Could you also provide here the amount of data in each bin, maybe by adding another y-axis for that?

Page 18, line 9: "*Here, it shows that the difference between S17 and W19 only occurs at the very lowest level*" - this cannot really be seen from Figure 12.

Page 19, Figure 11: Another horizontal axis with a km scale would be nice, because in the text you give the length of the flight path in kilometers.

Page 20, Figure 12: Different DWL lines are extremely difficult to see in both panels. Consider using different colors for the lines and rescaling of the figures.

Page 20, lines 16-18: "*The advection effects are most relevant at the lowest measurement heights where the spatial separation of lidar beams along the VAD cone Δy are small.*" This is true, but what could perhaps be also mentioned is that the advection speed increases with height because of the logarithmic wind profile.

Page 21, lines 1-3: "*dissipation rates of values smaller than $10^{-3}$ $m^2s^{-3}$ are underestimated by the lidars, likely because the small scale fluctuations that are carrying much of the energy in these cases, cannot be resolved any more.*" This can be true, but you should still check the method to retrieve dissipation rates from sonic anemometers as mentioned in the previous comments.

Page 21, lines 8-10: this information should be provided much earlier in the manuscript. This is not just a result but also the motivation to use research aircraft data in the first place.

---

## Author Comment (AC1) · 5 May 2020

**1   Author response**

We want to thank the two anonymous reviewers for their valuable feedback and valid points of criticism to our manuscript.

**1.1   RC1, General Comments**

1. *Through the comparison with sonics they are able to show a slight improvement when pulse averaging effects are considered (Fig 6).  The main novelty here*

[Figure]

*seems to be the advection correction, which they claim improves the retrievals, but the results are a bit underwhelming (see Fig. 6 and 7). I'm not at all convinced that the tiny improvements in the metrics (bias and correlation) are significant. I would be more inclined to conclude that the advection correction doesn't have a significant effect.*

The novelty of the paper is not only the advection correction. We re-evaluate a method introduced by Smalikho et al. and validate it in two different campaigns. The analysis method has been modified in ways that are relevant for practical implementation (smaller number of scans, VAD at higher elevation angle) and it was shown that these modifications are legit. The paper describes the first comparison of in-situ aircraft data with lidar turbulence retrievals above 800 m. We show that more validation of this kind will be necessary in future because lidar measurements in low turbulence regimes can significantly underestimate TKE and especially its dissipation rate.

The advection correction has only a small effect for low elevation VAD scans, which is a positive result as it shows that the Smalikho-method can be used with less restrictive advection filters, if the respective uncertainties are acceptable. Nevertheless, we do show an improvement of the TKE error which increases with higher wind speeds (Figure 8) especially at the 50-m level if the correction is applied. Even more importantly, the advection correction is highly effective if higher elevation VAD scans (here: 75°) are performed to retrieve TKE dissipation rate as is shown in Figs. 9 and 10.

**1.2   RC1, Specific Comments**

1. *Abstract: The first 3 or 4 sentences could be probably be reduced to a single sentence in favor of allowing for a more quantitative summary later in the abstract. As it stands, the abstract lacks sufficient substance. The author should incorporate more hard results from the comparisons with sonic anemometers.*

We believe that we need to explain the basic idea of the measurements with the introductory sentences even in the abstract but can still add more substance to it in the end. A modified version is given in the revised manuscript.

2. *page 4 line 3: Not everyone will know where Upper Silesia is (including myself until I looked it up), I suggest "...were installed in Upper Silesia, in southern Poland (or where ever), ..."*
We will add the country Poland in parantheses, the exact location is indicated on a map of Europe in Figure 2.

3. *page 4 line 15: change "...and were finally fixed..." to "...and were finally choosen..."*
Ok.

4. *Table 1. The Stream Line wavelength is 1.548 $\mu$m.*
From the lidar manufacturer Halo Photonics, we were only provided with the official value of 1.5 $\mu$m. If the reviewer can give us a reference for the value he suggests, we will very much like to correct it in the table. On the other hand, this value is not very important for the study and if serious doubts about this value persist, we would rather not mention it at all.

5. *Section 3.2: The author should include the equation for the measured azimuth structure function – since this is key for the dissipation rate retrieval methods.*
We will include the equation just before Eq. 8 in the revised manuscript:

$$D_L(\psi_l) = \langle [v_r(\theta) - v_r(\theta + \psi_l)]^2 \rangle$$
$$D_a(\psi_l) = D_L(\psi_l) - 2\sigma_e^2$$

6. *Equation 3: The condition that $\varphi = 35.3°$ should be made more explicit to prevent possible misused.*
The sentence introducing Equation 3 explicitly states that it is for the special case

of $\varphi = 35.3°$.

We add the suggested addition to the equation.

7. *Page 6, lines 4-6: The author states that the "TKE dissipation rate is estimated through a fit of the measured second-order structure function of horizontal velocity to the theoretical ..." This statement implies that the observations are adjusted to fit the model, when in fact it's the other way around, i.e. the model parameters are adjusted in order to fit the observations. Please rephrase.*

We apologize for the mistake in language and correct it in the revised manuscript.

8. *Page 6, lines 23-24: Similar to last comment. The author states that "A fit of the azimuth structure to the equation ..." again implies that the observations are being adjusted to fit the model, when in fact it's the other way around. Please rephrase.*

We apologize for the mistake in language and correct it in the revised manuscript.

9. *Page7, line 1: The author states that "Scanning with Doppler lidar in a VAD implies a volume averaging of radial velocities in longitudinal and transverse directions." Aside from the grammatical errors, this statement is not generally true because transverse averaging is not an issue for step-stair scans, only for continuous motion scans. The author should briefly mention the two different types of scans in their introduction. Also, the author should define what they mean by longitudinal and transverse (i.e. along the beam, and orthogonal to the beam).*

Velocity azimuth display (VAD) is a term from radar technology where continuous motions of the azimuth motor are the standard. Step-and-stare scans like Doppler-Beam-Swinging techniques are not considered in this manuscript. In any case, even step-and-stare scans with pulsed DWL have a transverse averaging effect due to the pulse-averaging over a certain accumulation time. We add to the manuscript that transverse averaging is regarded for scans with a continuous motion. We also define longitudinal and transverse in the revised manuscript.

10. *Page 7 line 5: change "... radial wind speed..." to " ...radial velocity...". Wind speed is a (positive) scalar, velocity is a vector. In this sentence your talking about the radial component of the velocity vector. "Radial wind speed" makes no sense.*
We change the wording in the revised manuscript.

11. *Page 7 starting at line 7: The discussion here is a bit disjointed and difficult to follow. Equations 5-7 should be listed after the sentence on line 5 (starting with "It is based on the decomposition..."). As it is, these equations are introduced without any corresponding text. One suggestion might be:*
*"In Smalikho and Banakh (2017), this method has been combined with the E89-method to yield TKE, and the momentum fluxes. It is based on the decomposition of radial velocity variance into its subcomponents, i.e.* $\sigma_L^2 = \sigma_a^2 + \sigma_e^2$
$\sigma_a^2 = \sigma_r^2 - \sigma_t^2$
$\sigma_r^2 = \sigma_L^2 + \sigma_t^2 - \sigma_e^2$ *where $\sigma_L^2$ is the lidar measured variance, $\sigma_a^2$ is the lidar measured variance without instrumental error, $\sigma_e^2$ is the instrumental error variance, and $\sigma_t^2$ is turbulent broadening of the lidar measurement. In Smalikho and Banakh (2017), all of these variances and corresponding structure functions are calculated for single azimuth angles and then averaged."*
We agree that the modifications make the text easier to follow and incorporate the changes in the revised manuscript.

12. *Page 7, line 17: Recommend changing "Substituting $\sigma_e^2$ in Eq. 7 with Eq. 8 yields:" to "Combining Eq. 7 with Eq. 8 yields:"*
We change the sentence in the revised manuscript according to the suggestion of the referee.

13. *Page 7, lines 18-23, including equations 9, 10 and 11: There is a dependence on the separation distance on the right side of equation 9 that presumably cancels such that the right side is effectively constant, i.e. independent of separation*

[Figure]

*distance. This is a subtle point that is not made by the author. Also, in equation 10, the author has substituted $\Psi_l$ with $\Psi_1$ without any explanation or justification. Please explain.*

We agree that an explanation is lacking here. Since the instrumental error $\sigma_e^2$ is assumed to be a constant offset of azimuth structure function $D_a(\psi_l)$ and the lidar measurement $D_L(\psi_l)$, $l$ can actually be chosen arbitrarily in the TKE equation. It is set to $l = 1$ because potential random errors like unstationary flow will be least effective for small separation angles.

14. *Page 8, line 10-11: The author states ". . .from VAD scans with other elevation angles as well." You should be a bit more specific here, since readers may not know what you mean by "other elevation angles." I assume you're referring to elevation angles different from 35.3°.*

We change the sentence to explicitly say "elevation angles different from 35.3°.

15. *Page 8, line 13-15: The author states "The value of l = 9 is chosen following the example of Smalikho and Banakh (2017) and corresponds to $l\Delta\theta = 9°$ as it was found to be suitable in all conditions in that study." The discussion up to this point had been fairly general. Now, suddenly the author is referring to a very specific VAD scan. The author should be a bit more specific as to which scan (and which experiment) they are referring to.*

Here we define the upper separation angle that will be used in the further manuscript and in all experiments. The separation angle is not referring to a specific VAD scan and in our opinion can be introduced here. We rephrase in the revised manuscript to state that this separation angle was found by Smalikho and Banakh (2017) to be a reasonable value in the ABL. It is illustrated by an example structure function in Figure 3.

16. *Page 8, line 24: change ". . .radial wind speeds. . ." to ". . .radial velocities. . ."*

All occurences of the term "radial wind speed" are replaced with "radial velocity".

17. *Page 8, line 28: change "...radial wind speeds..." to "...radial velocities..."*
    All occurences of the term "radial wind speed" are replaced with "radial velocity".

18. *Page 10, line 1: change "...radial wind speeds..." to "...radial velocities..."*
    All occurences of the term "radial wind speed" are replaced with "radial velocity".

19. *Page 10, line 3: change "...radial wind speed..." to "...radial velocitiy..."*
    All occurences of the term "radial wind speed" are replaced with "radial velocity".

20. *Page 10, line 6: Recommend changing "Since the mean of the radial wind speed fluctuations vr = 0 by definition, it is:" to something like "Since the mean of the radial velocity fluctuations is zero by definition, equation (20) becomes "*
    We change the sentence accordingly.

21. *Page 10 line 5: The author states "(here: g=360 for all azimuth angles)". The reference to a specific value of g here is a bit perplexing. Please explain.*
    In this study we work with VAD scans with $1°$ azimuthal resolution, which yields 360 values per scan. Since we introduce a general method here, we will remove this information at this point of the text.

22. *Equation 20: The summation is over j, but there is no dependence on j in the quantity being summed. Please explain.*
    This is a mistake. The summation is over the variable $m$ (index of the azimuth angle within one VAD scan).

23. *Page 10, lines 11-12: change "...radial wind speeds..." to "...radial velocities..."*
    All occurences of the term "radial wind speed" are replaced with "radial velocity".

24. *Page 10, line 16-17: The author states "Measured PDFs of the variables ... are fit to the model PDFs to obtain an estimation of the corresponding standard deviations $\sigma_1$, $\sigma_2$ and $\sigma_3$ and probability of bad estimates $P_1$, $P_2$ and $P_3$." This statement implies that the observations are fit to the model. In other words, the*

*observations are tweaked to get agreement with the model. That's certainly not what is happening. Please rephrase.*

We apologize for the confusion in language and rephrase in the revised manuscript.

25. *Section 3.2.2: It seems to me there is some slightly circular logic going on here. From what I gather, your fitting equation 22 to the measured PDFs to obtain estimates of the variance and the false-alarm probability. But since the real distributions aren't Gaussian, you end up computing the variance directly from the data. This begs the question as to why the variance was treated as an adjustable parameter in the first place. Why not just compute the variance from the data to begin with and then use equation 22 to estimate only the false alarm probability. What do the distributions look like? How good (or bad) are the fits?*

A first guess of the standard deviations is needed to find the $\pm 3.5\sigma$ region for the integral over the PDF. All the details of this method are given in Stephan et al. (2018). Since this method is not essential for this study, we do not want to expand too much on it in this manuscript. It is mostly relevant if better measurements in low-signal conditions are targeted.

26. *Section 3.2.3: In this section the author throws down a series of equations without adequate discussion. The authors need to do a better job explaining their line of reasoning.*

We thank the referee for their recommendation on improving the section and incorporate it in the revised manuscript.

27. *Appendix C, lines 17: The author introduces the quantity Xj (i.e. a 1D vector), and then in equation C1 it is indicated to be a 2D matrix, i.e. Xij. Please explain.*

$j$ is the subsample index and $i$ is the index for each element of the subsample. We clarify this in the revised manuscript.

28. *Appendix D: I find no mention of the "FSWF-retrieval" in the paper (did I miss it?). Please elaborate and provide relevant citations.*
Filtered sine-wave fitting is introduced with the corresponding reference in Section 3.2.1, but without giving the abbreviation FSWF. We add this in the revised manuscript.

**References**

Stephan, A., Wildmann, N., and Smalikho, I. N.: Spatiotemporal visualization of wind turbulence from measurements by a Windcube 200s lidar in the atmospheric boundary layer, Proc.SPIE, 10833, 10 833 – 10 833 – 10, 10.1117/12.2504468, 2018.

---

## Author Comment (AC2) · 5 May 2020

**1   Author response**

We want to thank the two anonymous reviewers for their valuable feedback and valid points of criticism to our manuscript.

**1.1   RC1, General Comments**

1. *The method to retrieve $\varepsilon$ from sonic anemometer (and research aircraft) measurements is based on averaging 2 min estimates of $\varepsilon$ over 30 min sampling*

[Figure]

*time. Taking the average means that the distribution of $\varepsilon_{2min}$ is assumed to be Gaussian, which is not necessarily true as the magnitude of $\varepsilon_{2min}$ may vary over several scales of magnitude. When the distribution is not Gaussian, the average will introduce a bias to the $\varepsilon_{30min}$ estimate. Therefore, authors should check the shape of the distribution of $\varepsilon_{2min}$ values during each 30 min period and choose an estimate for $\varepsilon_{30min}$ that is more representative for the distribution of $\varepsilon_{2min}$ values, such as the median of these values.*

We thank the referee for this comment which is very valid and should be considered in the evaluation of sonic anemometer data. An arithmetic mean of dissipation rates $\varepsilon$ is not the best solution given the exponential character of the variable. Instead of using the median as suggested, we however believe now that for this study it is more reasonable to calculate the structure function over the full half-hour period and estimate $\varepsilon$ from it. Muñoz-Esparza et al. (2018) calculated 2-minute periods because they wanted to see bursts of turbulence on shorter timescales. In our case we are however comparing to lidar measurements that are averaged over half-hour periods, so that a comparison can be best made with the same period for the calculation of the structure function of sonic anemometer data.

We did investigate the difference between median of 2-minute estimates, mean of 2-minute estimates and 30-minute estimates and can confirm that the referee is right that the 2-minute mean is skewed towards larger values compared to the the median approach. A systematic error can however also occur when the median underestimates the dissipation rate within the half-hour (see Fig. 1). We will not present these results in the revised manuscript because we believe that 30-minute stucture function estimates are the right choice for this analysis.

In any case, for all possible estimates of $\varepsilon$ from sonic anemometers, the differences are small enough to not change the conclusions that are made for the comparison to the lidar retrievals. In Fig. 2 below, we show the differences: a)-c) show the 30-minute structure function estimate, d)-f) show the mean of 2-minute structure function estimate (as in the discussion manuscript) and g)-j) show the median of 2-minute structure function estimate. The best correlations can be found with the 30-minute structure function estimate. The median estimate shows the highest bias for the S17 method, but the general conclusions remain the same.

**1.2 RC1, Specific Comments**

1. *It is not obvious to all readers that "DLR Cessna Grand Caravan 208B" is a research aircraft, please, add this to the abstract. Further, it is not clear why do you introduce the research aircraft data in this study as the title is about improvement of turbulence estimation using Doppler wind lidars. For validation purposes research aircraft data cannot be considered as robust as sonic anemometer data. In fact, the explanation for using aircraft measurements is provided only in Section 5 on lines 8-9 of page 21. This explanation should be given already in the abstract but also in the Introduction (Section 1) and maybe also in Section 3.3.*
We add the term "research aircraft" in the abstract explicitly and also explain that the research aircraft is a unique possibility to collect in-situ turbulence measurements above the heights that are in reach with sonic anemometers for example.

2. *Section 2, page 2, lines 33-34: "data from two different sites and sets of instruments": This is not clear: you use data from four DWLs (three in Upper Silesia, one at Falkenberg), from two sonic anemometers and from one research aircraft. Although you introduce the measurements in two Figures (Figures 1 and 2), it does not change the fact that you have several type of instruments and sites, not just two of each.*
We change the text to only state that measurements from two different experiments are analyzed. The details of the sites and the instrumentation is given in detail throughout the section: "In this study, data from two different experiments are analyzed. Both of the experiments and the instrumentation is introduced in

this section."

3. *Section 2.1., page 3, line 19: remove "and infrared gas analysers LI7500 (LiCor Inc.)" because you do not use this data.*
We remove the information about the gas analyzer in the text: "Continuous turbulence measurements (20 Hz sampling frequency) using sonic anemometer of type USA-1 (METEK GmbH) are performed at the 50m and 90m levels of the tower and have been used for validation purposes. The instruments are mounted at the tip of the booms pointing towards South."

4. *Section 2.2, the first two paragraphs on page 4: the description of the CoMet mission is too detailed and not relevant to the topic of this paper, as here the aim is not to investigate CO2 or methane but to develop DWL data processing methods. Please, provide here only such information that is relevant for the present study.*
We believe that the information about the CoMet campaign is relevant, because it puts the DWL measurement into context for the special issue to which this manuscript has been submitted. This manuscript is important for the CoMet research community, so that we want to make a connection to the overall project.

5. *Page 4, line 19: angle should read $35.5°$ not $35°$.*
Ok.

6. *Page 4, lines 20-21, 2 comments:*

   (a) *Check the tense of verbs to be consistent.*

   (b) *The acronyms of the Doppler lidars in Upper Silesia region are misleading: for the research aircraft you use the acronym "DLR" (e.g. in the abstract but also on line 8 of page 4) and for Doppler lidars you have introduced the acronym "DWL". Why do you introduce here another acronym for DWL, i.e., "DLR"? Please, use only one acronym for Doppler wind lidars throughout the paper.*

We change the naming of the lidars in the CoMet campaign to DWL#1-DWL#3 in the revised manuscript. The tense of the verb are all changed to past tense.

7. *Page4, lines 23-25: "a case study on 5 June 2017, on which D-FDLR was performing long straight and level legs between 800 m and 1600 m as indicated in the flight path in Fig. 2." This is the first time you introduce the research aircraft data and it is somewhat vague. Please, provide more information on why did you choose this data set, how did you select this specific period, what kind of instrumentation there was onboard, how accurate are the turbulence measurements from the aircraft in general, etc?*
More information with the relevant reference is given in this Section in the revised manuscript.

8. *Page 6, line 7: "the values are calculated for 2-minute intervals and then averaged over half-hour periods.". Include here information about the distribution of values calculated for 2-minute intervals, to show that the average is (or is not) a representative parameter for the population of values (see also the major comment).*
See above answer to major comment.

9. *Page 7, line 19: Should it read Eq.2 instead of Eq.3?*
It is actually Eq. 3, but the way this is written is confusing and we rephrase in the revised manuscript.

10. *Page 7, line 20: Why does $\Psi_l$ change to $\Psi_1$ here? Please, explain what it means that l=1?*
We agree that an explanation is lacking here. Since the instrumental error $\sigma_e^2$ is assumed to be a constant offset of azimuth structure function $D_a(\psi_l)$ and the lidar measurement $D_L(\psi_l)$, $l$ can actually be chosen arbitrarily in the TKE equation. It is set to $l = 1$ because potential random errors like unstationary flow will be least effective for small separation angles.

11. *Page 7, line 25: Typo: "Kolmogorov-Obhukov spectrum"*
    ok.

12. *Page 8, Equation (15): It is not clear how this Equation is derived from Eq. 13 and 8: what happens to $\sigma_e^2$ that was in Equation 8?*
    By taking the difference of $D_a(\psi_l) - D_a(\psi_1)$, $\sigma_e^2$ gets eliminated. We rephrase slightly to be clearer: "Using Eq. 13 and 8, $\varepsilon$ can be retrieved from $D_a(\psi_l) - D_a(\psi_1)$:"

13. *Page 9, line 8: "In (Smalikho and Banakh, 2017)" change to "In Smalikho and Banakh (2017)"*
    Ok.

14. *Page 10, Equation 20: There is no index j in the equation (which is included in the summation). Moreover, are there some parenthesis missing?*
    The index $j$ is a mistake, it should be $m$. There are no paranthesis missing.

15. *Page 12, Section 3.3: Again, it is not clear why research aircraft data is used: is it used a) because it gives more reliable results than DWLs and therefore can be used for validation of DWL data or b) is it used because it would be interesting to know how good the research aircraft data is compared to DWLs (and sonics)?*
    We consider turbulence measurements with flow probes on research aircraft a well-established method which provides more reliable measurements than a DWL since it does not depend on many assumptions except the Taylor's hypothesis.

16. *Page 14, Figure 5: It is not possible to see the dotted line in panel (a). Moreover, in the caption, could you provide the Equation numbers for the averaged variance and total variance methods in order to strengthen the link between the theory and the results.*
    Since the lines for the both methods lie almost on top of each other and the dotted line is thus hard to see we include a subplot showing the difference between the

two methods for the presented time series (see Fig. 3). The references to the equations are added in the revised manuscript.

17. *Page 14, line 5: "modified version W19 introduced in this study", maybe you should use acronym W20 for the method introduced in this paper?*
We changed the acronym throughout the manuscript.

18. *Page 15, Figure 6: Does the biases in (b) and (e) include all points or only those after the advection filter?*
All points except the grey points below the threshold are used. We will clarify in the caption. We think it is important to use all points to evaluate the effect of the advection correction.

19. *Page 17, Figure 8: Why there is an oscillatory pattern in TKE bias as a function of horizontal wind speed? The oscillatory pattern is more significant than the differences between the methods, and therefore it should be discussed. Could you also provide here the amount of data in each bin, maybe by adding another y-axis for that?*
There is no physical reason for an oscillatory pattern in TKE vs. horizontal wind speed. The shape of the curve is merely a specialty of the dataset where more values of certain bins fall in nighttime hours with lower TKE and thus also lower absolute error in TKE. We add the number of data points in the bin in Figure 8 of the manuscript as shown here in Fig. 4.

20. *Page 18, line 9: "Here, it shows that the difference between S17 and W19 only occurs at the very lowest level" - this cannot really be seen from Figure 12.*
It shows that for DLR1, there is a small difference between the two methods, but it is not very well visible. We change the discussion of this figure in the revised manuscript.

[Figure]

21. *Page 19, Figure 11: Another horizontal axis with a km scale would be nice, be-cause in the text you give the length of the flight path in kilometers.*
We change the plot to give the x-axis in kilometers instead.

22. *Page 20, Figure 12: Different DWL lines are extremely difficult to see in both panels. Consider using different colors for the lines and rescaling of the figures.*
We rescaled the figure and changed the color of the greyscale for the S17-method slightly for better readability of the plot.

23. *Page 20, lines 16-18: "The advection effects are most relevant at the lowest measurement heights where the spatial separation of lidar beams along the VAD cone $\Delta y$ are small." This is true, but what could perhaps be also mentioned is that the advection speed increases with height because of the logarithmic wind profile.*
We add this thought to the discussion. In our observation, the effect of increasing wind speed with height is however not as strong as the effect of a counteracting larger $\Delta y$.

24. *Page 21, lines 1-3: "dissipation rates of values smaller than $10^{-3} m^2 s^{-3}$ are un-derestimated by the lidars, likely because the small scale fluctuations that are carrying much of the energy in these cases, cannot be resolved any more." This can be true, but you should still check the method to retrieve dissipation rates from sonic anemometers as mentioned in the previous comments.*
As shown above we have evaluated the sonic anemometer retrieval and agree that there can be differences between different methods to obtain a representa-tive value of $\varepsilon$ within the half-hour period. However, the strong underestimation of lidar measurements below $10^{-3} m^2 s^{-3}$ is much more significant and found in any case.

25. *Page 21, lines 8-10: this information should be provided much earlier in the manuscript. This is not just a result but also the motivation to use research air-*

*craft data in the first place.*
More information on the motivation to use aircraft data is added to Section 2.2. in
the revised manuscript.

**References**

Muñoz-Esparza, D., Sharman, R. D., and Lundquist, J. K.: Turbulence Dissipation Rate in
the Atmospheric Boundary Layer: Observations and WRF Mesoscale Modeling during the
XPIA Field Campaign, Monthly Weather Review, 146, 351–371, 10.1175/MWR-D-17-0186.1,
2018.

[Figure]

[Figure]

**Fig. 1.** Comparison of different estimations of dissipation rate from sonic anemometers

[Figure]

**Fig. 2.** Scatter plot of lidar dissipation rate retrieval against sonic anemometer dissipation rate, calculated with different methods.

[Figure]

**Fig. 3.** Modified Figure 5 of the manuscript.

[Figure]

**Fig. 4.** Modified Figure 8 of the manuscript.

---

## Referee Report (RR1)

**Reviewer 1 comments on 1ˢᵗ draft are in blue**
**Author replies are in black**
**Reviewer 1 comments on 2ˢᵗ draft are in red**

RC1, General Comments

Through the comparison with sonics they are able to show a slight improvement when pulse averaging effects are considered (Fig 6). The main novelty here seems to be the advection correction, which they claim improves the retrievals, but the results are a bit underwhelming (see Fig. 6 and 7). I'm not at all convinced that the tiny improvements in the metrics (bias and correlation) are significant. I would be more inclined to conclude that the advection correction doesn't have a significant effect.

The novelty of the paper is not only the advection correction. We re-evaluate a method introduced by Smalikho et al. and validate it in two different campaigns. The analysis method has been modified in ways that are relevant for practical implementation (smaller number of scans, VAD at higher elevation angle) and it was shown that these modifications are legit. The paper describes the first comparison of in-situ aircraft data with lidar turbulence retrievals above 800 m. We show that more validation of this kind will be necessary in future because lidar measurements in low turbulence regimes can significantly underestimate TKE and especially its dissipation rate. The advection correction has only a small effect for low elevation VAD scans, which is a positive result as it shows that the Smalikho-method can be used with less restrictive advection filters, if the respective uncertainties are acceptable. Nevertheless, we do show an improvement of the TKE error which increases with higher wind speeds (Figure 8) especially at the 50-m level if the correction is applied. Even more importantly, the advection correction is highly effective if higher elevation VAD scans (here: 75_) are performed to retrieve TKE dissipation rate as is shown in Figs. 9 and 10.

The revised manuscript is improved, and I believed should be published with minor revisions. I believe the authors should be careful about claiming "the first comparison of in-situ aircraft data with lidar turbulence retrievals", as I point out in the comments below. There are statements to this effect in the abstract and the conclusions section. The term "turbulence retrieval" is too ambiguous. The author should be more specific, i.e. replace "turbulence retrieval" with "TKE and TKE dissipation rate retrieval."

RC1, Specific Comments

- Abstract: The first 3 or 4 sentences could be probably be reduced to a single sentence in favor of allowing for a more quantitative summary later in the abstract. As it stands, the abstract lacks sufficient substance. The author should incorporate more hard results from the comparisons with sonic anemometers.
    o We believe that we need to explain the basic idea of the measurements with the introductory sentences even in the abstract but can still add more substance to it in the end. A modified version is given in the revised manuscript.
    o The revised abstract provides a better and more quantitative summary of the results. The grammar (specifically in the abstract) is still a bit rough. For example, the second sentence states "DWL measurements extend beyond the observations with meteorological masts and are comparably flexible in their installation." I believe the thought being expressed here is that the DWLs provide wind measurements above the level of meteorological masts while being much easier and much less expensive to deploy.
- page 4 line 3: Not everyone will know where Upper Silesia is (including myself until I looked it up), I suggest ". . . were installed in Upper Silesia, in southern Poland (or where ever), . . . "
    o We will add the country Poland in parantheses, the exact location is indicated on a map of Europe in Figure 2.

- o   Thanks.
- page 4 line 15: change ". . . and were finally fixed. . . " to ". . . and were finally choosen. . . "
  - o   Ok.
  - o   Thanks.
- Table 1. The Stream Line wavelength is 1.548 _m.
  - o   From the lidar manufacturer Halo Photonics, we were only provided with the official value of 1.5 _m. If the reviewer can give us a reference for the value he suggests, we will very much like to correct it in the table. On the other hand, this value is not very important for the study and if serious doubts about this value persist, we would rather not mention it at all.
  - o   Fair enough. 1.5 is fine. Its not a big deal.

- Section 3.2: The author should include the equation for the measured azimuth structure function – since this is key for the dissipation rate retrieval methods.
  - o   We will include the equation just before Eq. 8 in the revised manuscript
  - o   Good. Thanks.

- Equation 3: The condition that ' = 35:3_ should be made more explicit to prevent possible misused.
  - o   The sentence introducing Equation 3 explicitly states that it is for the special case of ' = 35:3_. We add the suggested addition to the equation.
  - o   Good. Thanks.

- Page 6, lines 4-6: The author states that the "TKE dissipation rate is estimated through a fit of the measured second order structure function of horizontal velocity to the theoretical . . . " This statement implies that the observations are adjusted to fit the model, when in fact it's the other way around, i.e. the model parameters are adjusted in order to fit the observations. Please rephrase.
  - o   We apologize for the mistake in language and correct it in the revised manuscript.
  - o   Revision is better. Although I would suggest changing "…TKE dissipation rate is estimated through a fit of the theory of the longitudinal Kolmogorov-structure function" to "…TKE dissipation rate is estimated through a fit of the theoretical longitudinal Kolmogorov-structure function." I'm not sure about the hyphenation though.

- Page 6, lines 23-24: Similar to last comment. The author states that "A fit of the azimuth structure to the equation . . . " again implies that the observations are being adjusted to fit the model, when in fact it's the other way around. Please rephrase.
  - o   We apologize for the mistake in language and correct it in the revised manuscript.
  - o   Better. Although the statement should probably be broken down into two sentences. For example, "A fit of the equation … to the observed azimuth structure function yields estimates of epsilon. In equation (4) ck is the Kolmogorov constant…"

- Page7, line 1: The author states that "Scanning with Doppler lidar in a VAD implies a volume averaging of radial velocities in longitudinal and transverse directions." Aside from the grammatical errors, this statement is not generally true because transverse averaging is not an issue for step-stair scans, only for continuous motion scans. The author should briefly mention the two different types of scans in their introduction. Also, the author should define what they mean by longitudinal and transverse (i.e. along the beam, and orthogonal to the beam).
  - o   Velocity azimuth display (VAD) is a term from radar technology where continuous motions of the azimuth motor are the standard. Step-and-stare scans like Doppler-Beam-Swinging techniques are not considered in this manuscript. In any case, even step-and-stare scans with pulsed DWL have a transverse averaging effect due to the pulse-averaging over a certain accumulation time. We add to the manuscript that transverse averaging is regarded for scans with a continuous motion. We also define longitudinal and transverse in the revised manuscript.
  - o   Changes noted. Okay.

- Page 7 line 5: change ". . . radial wind speed. . . " to " . . . radial velocity. . . ". Wind speed is a (positive) scalar, velocity is a vector. In this sentence your talking about the radial component of the velocity vector. "Radial wind speed" makes no sense.
  - We change the wording in the revised manuscript.
  - Changes noted. Okay.

- Page 7 starting at line 7: The discussion here is a bit disjointed and difficult to follow. Equations 5-7 should be listed after the sentence on line 5 (starting with "It is based on the decomposition. . . "). As it is, these equations are introduced without any corresponding text. One suggestion might be: "In Smalikho and Banakh (2017), this method has been combined with the E89-method to yield TKE, and the momentum fluxes. It is based on the decomposition of radial velocity variance into its subcomponents, ...., where $\_{2L}$ is the lidar measured variance, $\_{2\,a}$ is the lidar measured variance without instrumental error, $\_{2\,e}$ is the instrumental error variance, and $\_{2t}$ is turbulent broadening of the lidar measurement. In Smalikho and Banakh (2017), all of these variances and corresponding structure functions are calculated for single azimuth angles and then averaged."
  - We agree that the modifications make the text easier to follow and incorporate the changes in the revised manuscript.
  - Reads much better now. In the revision I would suggest "…$\sigma_a{}^2$ **is** the lidar measured variance without instrumental error $\sigma_e{}^2$, and $\sigma_t{}^2$ is the turbulent broadening of the lidar measurement.

- Page 7, line 17: Recommend changing "Substituting $\_{2e}$ in Eq. 7 with Eq. 8 yields:" to "Combining Eq. 7 with Eq. 8 yields:"
  - We change the sentence in the revised manuscript according to the suggestion of the referee.
  - okay

- Page 7, lines 18-23, including equations 9, 10 and 11: There is a dependence on the separation distance on the right side of equation 9 that presumably cancels such that the right side is effectively constant, i.e. independent of separation distance. This is a subtle point that is not made by the author. Also, in equation 10, the author has substituted $l$ with $1$ without any explanation or justification. Please explain.
  - We agree that an explanation is lacking here. Since the instrumental error $\_{25\,e}$ is assumed to be a constant offset of azimuth structure function $D_a(\,l)$ and the lidar measurement $D_L(\,l)$, $l$ can actually be chosen arbitrarily in the TKE equation. It is set to $l = 1$ because potential random errors like unstationary flow will be least effective for small separation angles.
  - Better. but I would suggest "…random errors will be smaller for small separation angles."

- Page 8, line 10-11: The author states ". . . from VAD scans with other elevation angles as well." You should be a bit more specific here, since readers may not know what you mean by "other elevation angles." I assume you're referring to elevation angles different from 35.3_.
  - We change the sentence to explicitly say "elevation angles different from 35.3_.
  - Good. Thanks.

- Page 8, line 13-15: The author states "The value of $l = 9$ is chosen following the example of Smalikho and Banakh (2017) and corresponds to $l\_\_ = 9\_$ as it was found to be suitable in all conditions in that study." The discussion up to this point had been fairly general. Now, suddenly the author is referring to a very specific VAD scan. The author should be a bit more specific as to which scan (and which experiment) they are referring to.
  - Here we define the upper separation angle that will be used in the further manuscript and in all experiments. The separation angle is not referring to a specific VAD scan and in our opinion can be introduced here. We rephrase in the revised manuscript to state that this separation angle

was found by Smalikho and Banakh (2017) to be a reasonable value in the ABL. It is illustrated by an example structure function in Figure 3.
- o okay

- Page 8, line 24: change ". . . radial wind speeds. . . " to ". . . radial velocities. . . "
  - o All occurences of the term "radial wind speed" are replaced with "radial velocity".
  - o okay

- Page 8, line 28: change ". . . radial wind speeds. . . " to ". . . radial velocities. . . "
  - o All occurences of the term "radial wind speed" are replaced with "radial velocity".
  - o oaky

- Page 10, line 1: change ". . . radial wind speeds. . . " to ". . . radial velocities. . . "
  - o All occurences of the term "radial wind speed" are replaced with "radial velocity".
  - o okay

- Page 10, line 3: change ". . . radial wind speed. . . " to ". . . radial velocitiy. . . "
  - o All occurences of the term "radial wind speed" are replaced with "radial velocity".
  - o okay

- Page 10, line 6: Recommend changing "Since the mean of the radial wind speed fluctuations vr = 0 by definition, it is:" to something like "Since the mean of the radial velocity fluctuations is zero by definition, equation (20) becomes "
  - o We change the sentence accordingly.
  - o okay

- Page 10 line 5: The author states "(here: g=360 for all azimuth angles)". The reference to a specific value of g here is a bit perplexing. Please explain.
  - o In this study we work with VAD scans with 1_ azimuthal resolution, which yields 360 values per scan. Since we introduce a general method here, we will remove this information at this point of the text.
  - o okay

- Equation 20: The summation is over j, but there is no dependence on j in the quantity being summed. Please explain.
  - o This is a mistake. The summation is over the variable m (index of the azimuth angle within one VAD scan).
  - o Good. The equation makes sense now.

- Page 10, lines 11-12: change 5 ". . . radial wind speeds. . . " to ". . . radial velocities. . . "
  - o All occurences of the term "radial wind speed" are replaced with "radial velocity".
  - o Okay

- Page 10, line 16-17: The author states "Measured PDFs of the variables . . . are fit to the model PDFs to obtain an estimation of the corresponding standard deviations $\_1$, $\_2$ and $\_3$ and probability of bad estimates $P_1$, $P_2$ and $P_3$." This statement implies that the observations are fit to the model. In other words, the observations are tweaked to get agreement with the model. That's certainly not what is happening. Please rephrase.
  - o We apologize for the confusion in language and rephrase in the revised manuscript.
  - o The revision is better, but there is still a little room for improvement. The author states "the best-fit model PDFs are found to to obtain the corresponding standard deviations…" I think what

the author means is that Equation (23) is fit to the observed distributions of …, …, and … by adjusting the values of sigma and P.

- Section 3.2.2: It seems to me there is some slightly circular logic going on here. From what I gather, your fitting equation to the measured PDFs to obtain estimates of the variance and the false-alarm probability. But since the real distributions aren't Gaussian, you end up computing the variance directly from the data. This begs the question as to why the variance was treated as an adjustable parameter in the first place. Why not just compute the variance from the data to begin with and then use equation 22 to estimate only the false alarm probability. What do the distributions look like? How good (or bad) are the fits?
  - A first guess of the standard deviations is needed to find the _3:5_ region for the integral over the PDF. All the details of this method are given in Stephan et al. (2018). Since this method is not essential for this study, we do not want to expand too much on it in this manuscript. It is mostly relevant if better measurements in low-signal conditions are targeted.
  - No need to expand the discussion, but you should clarify the point that a first guess estimate of the variance is obtained directly from the data (is that true?). Since the true distribution is not Gaussian the final variance estimate is obtained by integrating equation (23) in the range from +/- 3.5*sigma, according to Stephan et al. (2018).

- Section 3.2.3: In this section the author throws down a series of equations without adequate discussion. The authors need to do a better job explaining their line of reasoning.
  - We thank the referee for their recommendation on improving the section and incorporate it in the revised manuscript.
  - The revision does a much better at explaining the equations.

- Appendix C, lines 17: The author introduces the quantity Xj (i.e. a 1D vector), and then in equation C1 it is indicated to be a 2D matrix, i.e. Xij. Please explain.
  - j is the subsample index and i is the index for each element of the subsample. We clarify this in the revised manuscript.
  - okay

- Appendix D: I find no mention of the "FSWF-retrieval" in the paper (did I miss it?). Please elaborate and provide relevant citations.
  - Filtered sine-wave fitting is introduced with the corresponding reference in Section 3.2.1, but without giving the abbreviation FSWF. We add this in the revised manuscript.
  - Good, that helps. I would also suggest adding the reference to Smalikho (2003) in the appendix where you mention the FSWF-retrieval method.

**New Comments**

Abstract

The author states: "For the first time, the DWL VAD turbulence retrievals are compared to in-situ measurements of a research aircraft…" I would be careful here. We have published comparisons of vertical velocity distributions measured from Doppler lidar to those measured on a research aircraft (e.g. "Overview of the HI-SCALE Field Campaign: A New Perspective on Shallow Convective Clouds", BAMS, by J. Fast, and there are probably other examples). Technically that could be viewed a "turbulence retrieval." The author should be more specific here, e.g. "For the first time, retrievals of TKE and TKE dissipation rate derived from DWL VAD data are compared to in-situ measurements from a research aircraft…"

Conclusions section

In the revised manuscript the author states…
 "The MOL-RAO experiment allowed us to show that methods which do not account for the lidar volume averaging effect underestimate turbulence compared to sonic anemometers at 50 m and 90 m systematically. This has been shown for the first time with such a big dataset. The S17-method tackles this problem, …"

In the first sentence the use of the term "turbulence" is ambiguous. The author should be more specific here (i.e. TKE and epsilon).  Also, the author should remove "systematically" from the first sentence.

The second sentence is a little perplexing. I'm not sure if the author is suggesting that this is the first time anyone has observed the volume averaging effect or if this is the first time that these four different retrieval methods have been used on the same dataset, or if this is first time that the methods have been applied to the 50 and 90-m levels, or if this is the first time the method has been applied to "…such a big dataset", or something else. Why does it matter how big the dataset is?  This statement attempts to point out how this work is novel, but the meaning is obscure. Please clarify.
In the last sentence, I would suggest changing "tackles" to "handles" or "attempts to handle"

---

## Author Response (AR2)

**Towards improved turbulence estimation with Doppler wind lidar VAD scans**

Norman Wildmann[1], Eileen Päschke[2], Anke Roiger[1], and Christian Mallaun[3]

[1]Deutsches Zentrum für Luft- und Raumfahrt e.V., Institut für Physik der Atmosphäre, Oberpfaffenhofen, Germany
[2]DWD, Meteorologisches Observatorium Lindenberg - Richard-Aßmann-Observatorium, Lindenberg, Germany
[3]Deutsches Zentrum für Luft- und Raumfahrt e.V., Flugexperimente, Oberpfaffenhofen, Germany

**Correspondence:** Norman Wildmann (norman.wildmann@dlr.de)

**1 Review response**

We want to thank the two anonymous reviewers for their valuable feedback and valid points of criticism to our manuscript.

**1.1 Review Comment 1**

no futher comments

5 ### 1.2 Review Comment 2

**1.3 RC2, General Comments**

1. *The revised manuscript is improved, and I believed should be published with minor revisions. I believe the authors should be careful about claiming "the first comparison of in-situ aircraft data with lidar turbulence retrievals", as I point out in the comments below. There are statements to this effect in the abstract and the conclusions section. The term "turbulence*
10 *retrieval" is too ambiguous. The author should be more specific, i.e. replace "turbulence retrieval" with "TKE and TKE dissipation rate retrieval."*

   We will be more specific with the turbulence retrieval and refer to "ground-based retrievals of TKE and its dissipationn rate with the VAD method".

**1.4 RC2, Specific Comments**

15 1. *The revised abstract provides a better and more quantitative summary of the results. The grammar (specifically in the abstract) is still a bit rough. For example, the second sentence states "DWL measurements extend beyond the observations with meteorological masts and are comparably flexible in their installation." I believe the thought being expressed here is that the DWLs provide wind measurements above the level of meteorological masts while being much easier and much less expensive to deploy.*

20 We apologize for our English grammar, and rephrase according to the suggestion in the revised manuscript.

2. *Revision is better. Although I would suggest changing "...TKE dissipation rate is estimated through a fit of the theory of the longitudinal Kolmogorov-structure function" to "...TKE dissipation rate is estimated through a fit of the theoretical longitudinal Kolmogorov-structure function." I'm not sure about the hyphenation though.*
Changed.

3. *Although the statement should probably be broken down into two sentences. For example, "A fit of the equation ... to the observed azimuth structure function yields estimates of epsilon. In equation (4) ck is the Kolmogorov constant..."*
Changed.

4. *Reads much better now. In the revision I would suggest "...$\sigma_a^2$ is the lidar measured variance without instrumental error $\sigma_e^2$, and $\sigma_t^2$ is the turbulent broadening of the lidar measurement.*
Changed.

5. *Better. but I would suggest "... random errors will be smaller for small separation angles."*
Changed.

6. *The revision is better, but there is still a little room for improvement. The author states "the best-fit model PDFs are found to to obtain the corresponding standard deviations..." I think whatthe author means is that Equation (23) is fit to the observed distributions of ..., ..., and ... by adjusting the values of sigma and P.*
Changed.

7. *No need to expand the discussion, but you should clarify the point that a first guess estimate of the variance is obtained directly from the data (is that true?). Since the true distribution is not Gaussian the final variance estimate is obtained by integrating equation (23) in the range from +/- 3.5*sigma, according to Stephan et al. (2018).*
Changed.

8. *I would also suggest adding the reference to Smalikho (2003) in the appendix where you mention the FSWF-retrieval method.*
Done.

9. *The author states: "For the first time, the DWL VAD turbulence retrievals are compared to in-situ measurements of a research aircraft..." I would be careful here. We have published comparisons of vertical velocity distributions measured from Doppler lidar to those measured on a research aircraft (e.g. "Overview of the HI-SCALE Field Campaign: A New Perspective on Shallow Convective Clouds", BAMS, by J. Fast, and there are probably other examples). Technically that could be viewed a "turbulence retrieval." The author should be more specific here, e.g. "For the first time, retrievals of TKE and TKE dissipation rate derived from DWL VAD data are compared to in-situ measurements from a research aircraft..."*
As stated above we are more specific in the revised manuscript, changing the term to "ground-based DWL VAD retrievals

of TKE and its dissipation rate". However, we can not agree that the vertical velocity distributions that are shown in the the reference should be called a "turbulence retrieval".

10. *In the first sentence the use of the term "turbulence" is ambiguous. The author should be more specific here (i.e. TKE and epsilon). Also, the author should remove "systematically" from the first sentence.*
We do not think that the term turbulence is "ambiguous", but to be very explicit, we change the sentence according to the recommendation.

11. *The second sentence is a little perplexing. I'm not sure if the author is suggesting that this is the first time anyone has observed the volume averaging effect or if this is the first time that these four different retrieval methods have been used on the same dataset, or if this is first time that the methods have been applied to the 50 and 90-m levels, or if this is the first time the method has been applied to "... such a big dataset", or something else. Why does it matter how big the dataset is? This statement attempts to point out how this work is novel, but the meaning is obscure. Please clarify.*
We did not suggest that the well-known volume averaging effect has not been observed before. To be less "perplexing" and "obscure", we rephrase this statement and expand upon it: "The MOL-RAO experiment allowed us to investigate and validate the methods with a database of 30 days containing a broad variety of atmospheric conditions, with wind speeds ranging from $0..12 \, \mathrm{m\,s^{-1}}$ and TKE from $0..5 \, \mathrm{m^2 s^{-2}}$. This goes beyond short-term studies of only few days and specific atmospheric conditions that had been investigated by **?**. Furthermore, two sonic anemometers allowed to investigate the dependence of the retrieval on the measurement height."

12. *In the last sentence, I would suggest changing "tackles" to "handles" or "attempts to handle"*
Done.

**2 Relevant changes to the manuscript**

We list here the relevant changes to the manuscript:

1. Abstract:

   – Text modifications in response to referee comments.

2. Section 3:

   – Text and equation modifications in response to referee comments.

3. Conclusions:

   – Text modifications in response to referee comments.

   – Added a reference to a recent study about turbulence anisotropy.

4. Appendix:

   – Text modifications in response to referee comments.

**References**

[revised manuscript text omitted]